# Whole-genome resequencing reveals world-wide ancestry and adaptive introgression events of domesticated cattle in East Asia

Ningbo Chen et al.[#]

Cattle domestication and the complex histories of East Asian cattle breeds warrant further investigation. Through analysing the genomes of 49 modern breeds and eight East Asian ancient samples, worldwide cattle are consistently classified into five continental groups based on Y-chromosome haplotypes and autosomal variants. We find that East Asian cattle populations are mainly composed of three distinct ancestries, including an earlier East Asian taurine ancestry that reached China at least ~3.9 kya, a later introduced Eurasian taurine ancestry, and a novel Chinese indicine ancestry that diverged from Indian indicine approximately 36.6–49.6 kya. We also report historic introgression events that helped domestic cattle from southern China and the Tibetan Plateau achieve rapid adaptation by acquiring ~2.93% and ~1.22% of their genomes from banteng and yak, respectively. Our findings provide new insights into the evolutionary history of cattle and the importance of introgression in adaptation of cattle to new environmental challenges in East Asia.

The domestication of cattle from wild aurochs (*Bos primigenius*) was one of the most significant achievements of the Neolithic period. By supplying meat, leather, and draught force for ploughing and transportation, cattle became the most important livestock in East Asian agricultural society. Today, cattle are still an important domesticated animal resource in East Asia. Fifty-three indigenous cattle breeds with ~100 million animals are observed in China[1]. Modern cattle breeds probably descended from multiple domestication events of wild aurochs present in different geographic areas ~10,000 years before present (YBP)[2,3]. Two primary domestication centres in the Near East and the Indus Valley resulted in humpless taurine (*Bos taurus*) and humped indicine (*Bos indicus*) cattle, respectively[3]. Generally, indicine cattle can withstand high temperatures compared with taurine breeds. Population analyses based on genomic single-nucleotide polymorphism (SNP) data revealed three major groups of cattle, including Asian indicine, Eurasian taurine, and African taurine, and recovered the historical migratory routes of cattle from their centres of origins across the world[4–7]. However, the whole-genome diversity of cattle from East Asia has not been investigated in depth.

Although several local domestication events have been inferred[2,3], paternal and maternal lineage analyses are consistent with East Asian domestic cattle sharing origins with other taurine and indicine backgrounds[8–12]. Archaeological evidence suggested that *B. taurus* may have been imported into East Asia from West Asia during the late Neolithic period (between 5000 and 4000 YBP)[8,13] and *B. indicus* may have dispersed from the Indian subcontinent to East Asia at a later stage, from 3500 to 2500 YBP[14,15], which led the hybridisation between taurine and indicine cattle in Central China[10,11]. Based on genomic SNP array data, previous studies provided evidence of introgression within the genus *Bos*, such as *Bos javanicus* (banteng) introgression into Chinese Hainan cattle and bovine introgression into Mongolian yak in East Asia[6,16]. All these events contributed to the complex histories of East Asian cattle.

Here we analyse the genomes of 260 present-day and eight ancient cattle with the goal of tracing the ancestry components

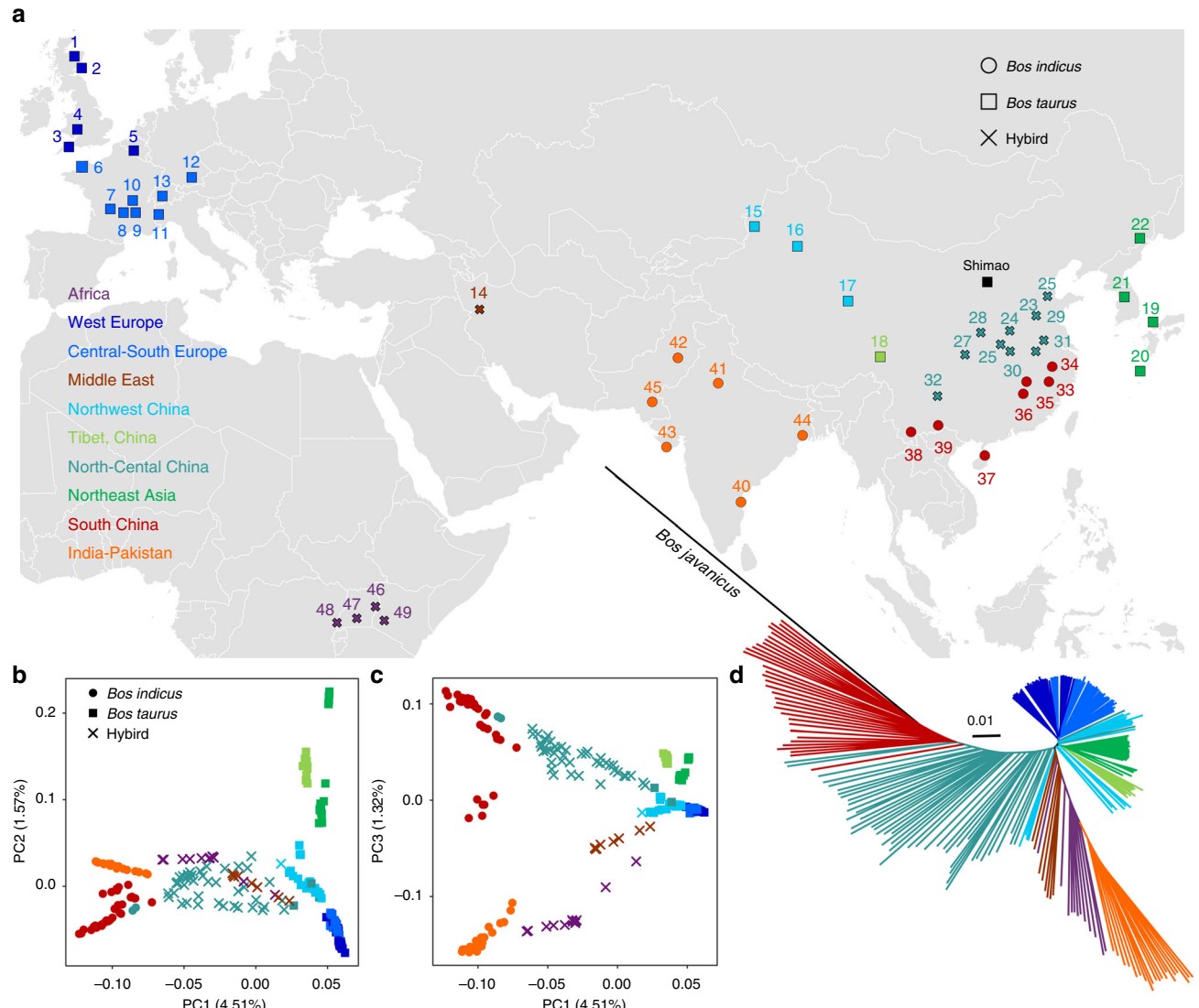

**Fig. 1** Population structuring and relationship of East Asian cattle. **a** Geographic map indicating the origins of the cattle breeds in this study. Breed name associated with serial number is listed in Fig. 2a. Principal component analysis (PCA) showing PC1 against PC2 (**b**) and PC1 against PC3 (**c**). **d** A neighbour-joining phylogenetic tree constructed using whole-genome SNP data. The scale bar represents pairwise distances between different individuals. Colours reflect the geographic regions of sampling

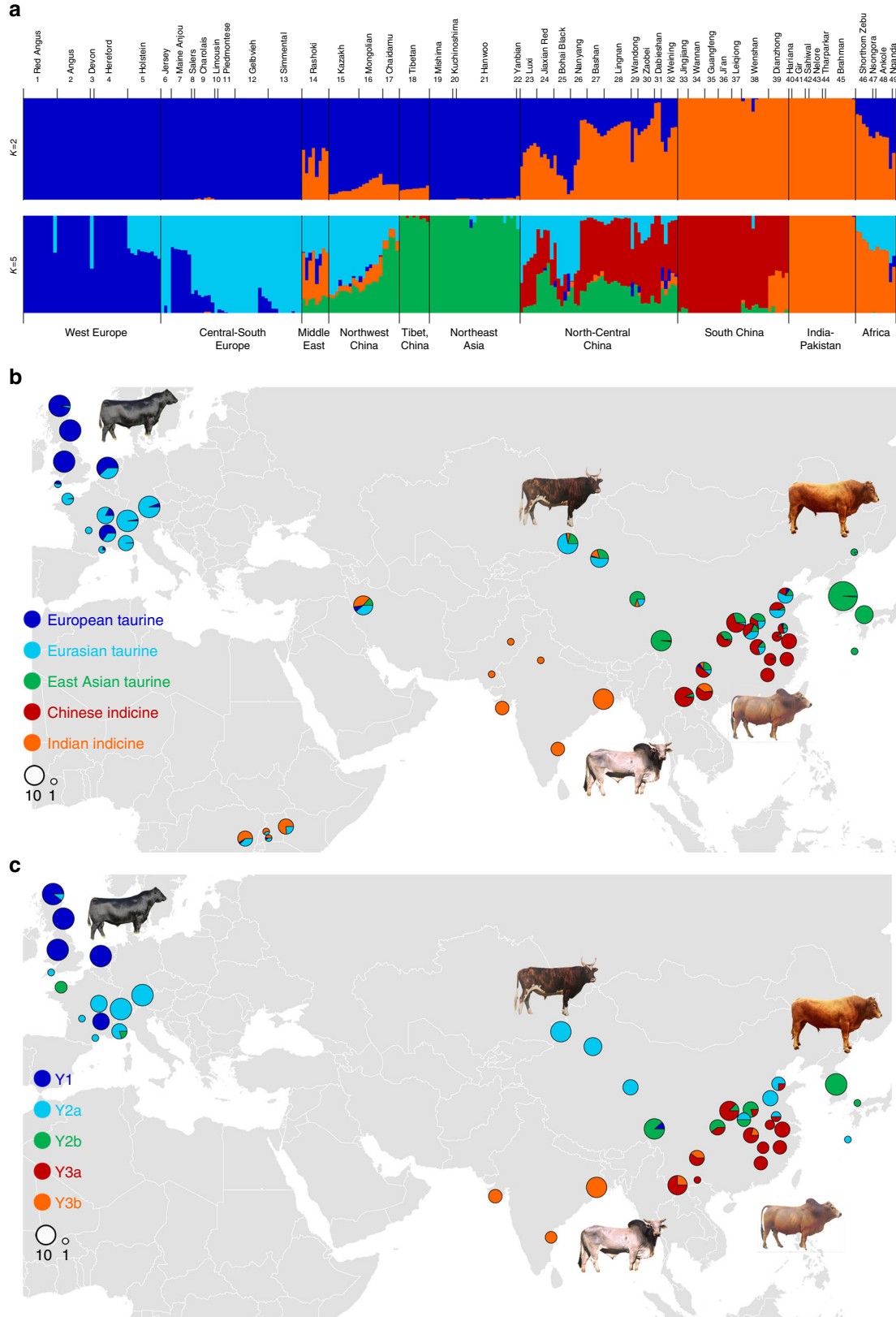

**Fig. 2** Autosomal and Y-chromosome evidence for the origin of modern cattle. **a** Model-based clustering of cattle breeds using ADMIXTURE with $K = 5$. Breeds are arranged by geographic regions and labelled with serial numbers. **b** Spatial distribution of five ADMIXTURE coefficients labelled according to geographical maxima. **c** Y haplogroup distribution of modern cattle. The size of each circle is proportional to the number of samples per breed. Five breeds and their corresponding colour codes are selected to represent the five types of ancestry and haplogroup

and recovering the historical migration and introgression processes that formed the currently observed diversity of East Asian cattle in this region.

## Results

**Genome resequencing**. A total of 111 modern animals representing 22 geographically diverse breeds in China and 3 indicine cattle representing 3 Indian breeds were selected for genome resequencing (Fig. 1a). To place these individuals into a global context, we also combined our data with available whole-genome resequencing data for 146 individuals from 24 breeds, giving a total of 260 individuals (Supplementary Table 1). Genome resequencing achieved an average depth of 12.72×. Of these animals, 213 bulls were used for a Y-chromosome SNP analysis. All animals were assigned to 10 geographical regions following their sources: Northeast Asia; Northwest China; North-Central China; Tibetan Plateau; South China; India-Pakistan; the Middle East; Africa; West Europe; and Central-South Europe (Supplementary Table 2). Gir, Brahman and Nelore, which were imported from India to the Americas approximately 200 years ago[17], were also used to represent Indian indicine cattle in this study (Supplementary Note 1 and Supplementary Table 3).

**Population genetic structure using autosomal variants**. Principal component analysis (PCA) demonstrated a clear genetic structure with samples from each geographical region clustering together (Supplementary Note 2). The first component was driven by difference between indicine and taurine cattle (Fig. 1b, c, and Supplementary Table 4). Within *B. taurus*, a separation was found between European and East Asian taurine cattle along the second component (Fig. 1b, Supplementary Fig. 1 and Supplementary Table 5). Within *B. indicus*, a clear partitioning was observed between cattle from India and South China (Fig. 1c, Supplementary Fig. 2 and Supplementary Table 6). The same population affinities were recovered in phylogenetic trees constructed by the neighbour-joining (NJ) and maximum-likelihood (ML) methods (Fig. 1d, Supplementary Fig. 4 and Supplementary Data 1). ADMIXTURE analysis also recapitulated these findings. When $K = 5$, we observed five geographically distributed ancestral components labelled: European taurine; Eurasian taurine; East Asian taurine; Chinese indicine; and Indian indicine (Fig. 2a, b, Supplementary Fig. 5 and Supplementary Table 7). European taurine ancestry was observed in West European breeds, such as Hereford and Angus. Eurasian taurine ancestry was identified at the highest frequency in both Central-South European breeds (Piedmontese, Gelbvieh, Limousin and Simmental) and Northwest Chinese breeds (Kazakh and Mongolian). We noticed that Tibetan and Northeast Asian breeds from two separate geographical regions formed a distinct East Asian taurine group. Within *B. indicus*, Indian indicine cattle (Hariana, Sahiwal, Tharparkar, Gir, Brahman and Nelore) were clearly separated from Chinese indicine cattle (Leiqiong, Jingjiang, Guangfeng, Ji'an and Wannan). The Dianzhong cattle from Yunnan in South China was composed of crosses with Indian-Chinese indicine genotypes. The Chinese indicine ancestry was observed at the highest frequency in South China and progressively decreased northward. Cattle breeds from other regions (the Middle East, Africa, Northwest China, North-Central China and Southwest China) showed evidence of hybridisation between *B. taurus* and *B. indicus* (Fig. 2a, b).

**Paternal and maternal phylogenetic analyses**. Both Y-chromosome and mitochondrial DNA (mtDNA) haplotypes represent strong foci in the investigations of prehistoric livestock. Here, after quality control and filtering (Supplementary Note 3),

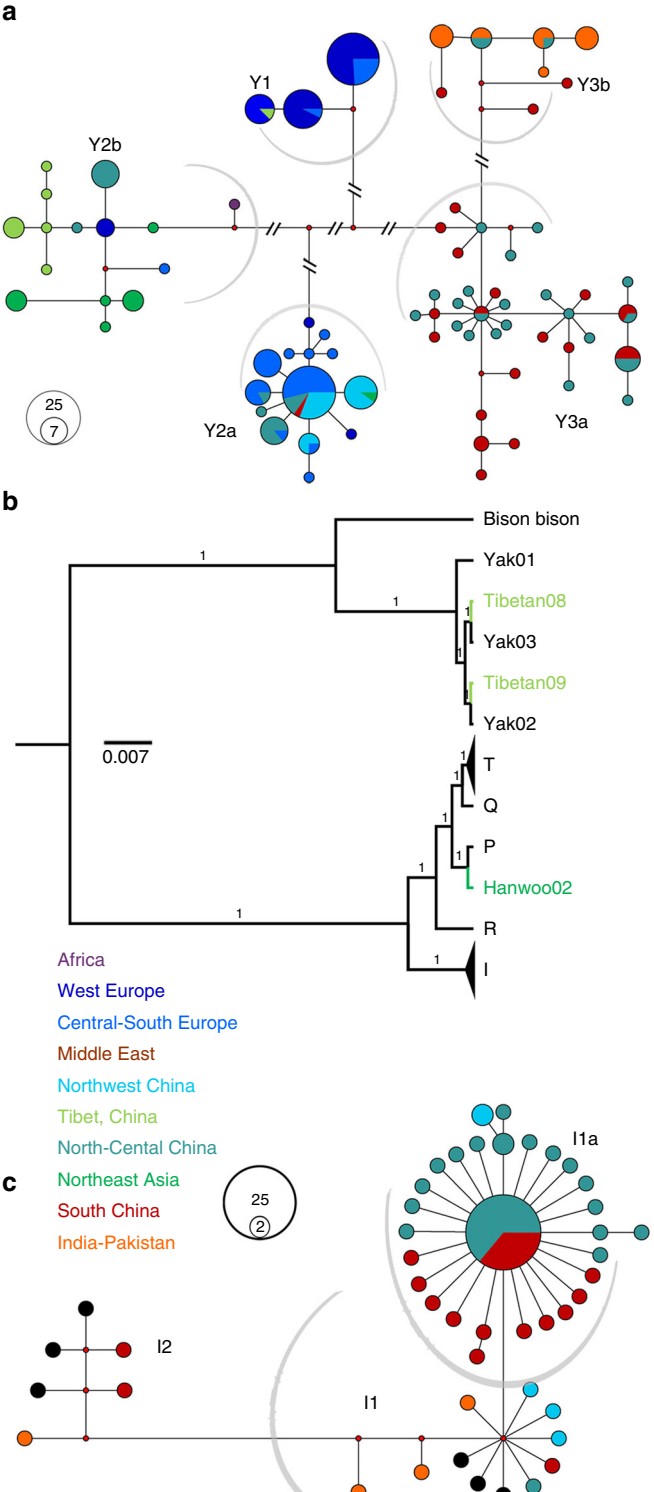

**Fig. 3** Y-chromosome haplogroup and phylogeny of complete mitogenomes from modern cattle. **a** Median-joining (MJ) network of Y-chromosome haplotypes using 745 SNPs. **b** Rooted maximum-likelihood phylogenetic tree detailing the relationships among all available bovine haplogroups of I, P, Q, R and T mitogenomes, an American bison mitogenome and three yak mitogenomes. **c** MJ network of mitogenomes from *Bos indicus*. The black line and circle in **b** and **c** represent published reference sequences, respectively

we used 745 SNPs in the X-degenerate region on Y chromosome[18] to construct a haplotype network (Fig. 3a and Supplementary Data 2) and rooted haplotype trees (Supplementary Fig. 7). Similar to earlier analyses that used several loci, three common Y haplogroups (Y1 and Y2 for taurine, and Y3 for indicine cattle) emerged[19]. Additionally, two distinct sub-haplogroups were resolved within each of the Y2 (Y2a and Y2b) and Y3 (Y3a and Y3b) haplogroups. Three *B. taurus*

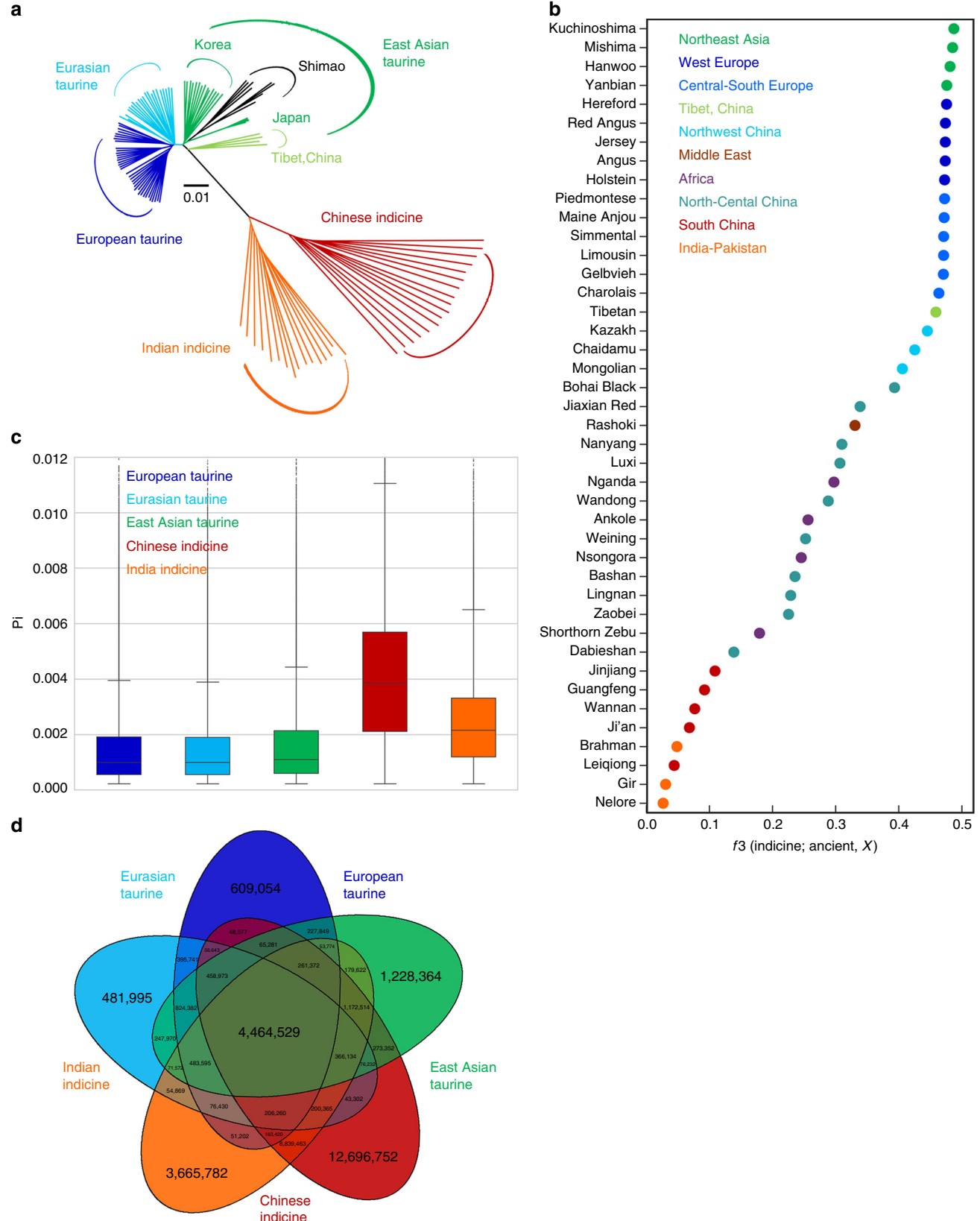

haplogroups were found in the European breeds. The sub-haplogroup Y2a was shared among Northwest Chinese breeds (Kazakh, Mongolian and Chaidamu) in East Asia while the sub-haplogroup Y2b dominated the cattle of Tibetan Plateau and Northeast Asia. The Y3 haplogroup was unique to *B. indicus*. The sub-haplogroup Y3a dominated the cattle from South China, whereas the sub-haplotype Y3b was mainly carried by three *B. indicus* breeds of Indian origin (Fig. 3a). North-Central China breeds admixed with Y2a, Y2b and Y3a (Figs. 2c and 3a).

We also inferred maternal lineages using complete mitogenomes. Modern taurine cattle exhibited the T haplogroup and several additional minor haplogroups (Q, P and R), whereas indicine cattle only carried the I haplogroup[20]. The 147 assembled mitogenomes presented >100-fold coverage, including 107 Chinese cattle, 12 African cattle, 11 European cattle, 7 Iranian cattle, 6 Korean cattle and 4 indicine cattle, were aligned with 24 reference mitogenomes (Supplementary Note 3 and Supplementary Table 8). Phylogenetic analyses showed that all East Asian taurine cattle were characterised by the T haplogroup except for a Korean Hanwoo, which belonged to the P haplogroup typical of European aurochs, and 2 Tibetan cattle, which carried yak haplogroups (Fig. 3b and Supplementary Fig. 8), suggesting the recruitment of aurochs and yak matrilines into East Asian cattle. The distribution of taurine mitogenome haplogroups was congruent with a previous study[21]. East Asian taurine cattle mainly belonged to the T3 and T2 haplogroups, the Middle East cattle belonged to the T2 haplogroup, African cattle belonged to the T1 haplogroup, and only 1 Tibetan individual belonged to the T4 haplogroup. Within the *B. indicus* lineage, Chinese indicine primarily belonged to I1a, a new sub-haplogroup within I1 (Fig. 3c and Supplementary Fig. 8) that diverges in a star-like fashion, suggesting its rapid population expansion from a single founder sequence.

**Ancient variation and sequential migration of cattle in East Asia.** DNA was successfully extracted from fossil cattle specimens excavated in the late Neolithic Shimao site in northern China (3975–3835 cal BP) (Fig. 1a, Supplementary Note 4, Supplementary Fig. 9 and Supplementary Table 9). Low-coverage genome sequences (ranging from 0.01× to 2.63× coverage) of eight ancient cattle were obtained (Supplementary Fig. 10 and Supplementary Table 10). To investigate the relationship of Shimao specimens to present-day East Asian cattle, we selected "core" groups based on the genetic structure analysis using ADMIXTURE and a phylogenetic analysis using outgroup-*f*3 statistics (Supplementary Note 4 and Supplementary Tables 11-15). The NJ tree revealed that the ancient samples showed the closest affinity for Korean Hanwoo and Japanese cattle (Fig. 4a). The outgroup-*f*3 statistics and *D* statistics also confirmed that the ancient cattle shared the most derived polymorphisms with Northeast Asian cattle and Japanese cattle, respectively (Fig. 4b, Supplementary Fig. 11 and Supplementary Table 16). Tibetan cattle were perhaps subjected to a stronger drift after the separation, which distorted their outgroup-*f*3 value (Fig. 4b). Combined with the distribution of paternal lineages (Figs. 2c and 3a), this finding suggests that two migration phases might have

occurred in the history of East Asian *B. taurus*. We speculate that the earliest cattle population (East Asian taurine) might be introduced before ~3900 YBP, whereas an exotic introgression (Eurasian taurine), which is now prevalent in the middle part of China, resulted from the second migration event.

**Possible *B. javanicus* introgression into Chinese indicine cattle.** The Chinese indicine genomes clustered separately from those of Indian indicine cattle and showed the highest nucleotide diversity among all geographic groups (Fig. 4c and Supplementary Fig. 12). Intriguingly, a total of 12,696,752 SNPs were found to be specific to Chinese indicine cattle, and this value was more than nine million higher than that of any other cattle group (Fig. 4d). This increased unique diversity could be influenced by particular (but unknown) historical demography such as population expansion but the scale of this unique diversity suggests hybridisation with or introgression from different bovine species. Thus, we used whole-genome resequencing data of seven Bovini species to determine the ancestral states for Chinese indicine-specific SNPs (Supplementary Note 5 and Supplementary Tables 1 and 2). The comparisons showed that Chinese indicine cattle and *B. javanicus* shared the most derived alleles (~4.7 M SNPs), followed by *B. taurus* and *Bos frontalis* (gayal) (Supplementary Table 17), suggesting interspecies introgression in the history of Chinese indicine cattle. Therefore, we used a combination of analyses, including *f*3 statistics, *D* statistics and Treemix, to test this introgression hypothesis. We ran *f*3 statistics on population triples using Chinese indicine cattle as a target (admixture population) and Indian indicine cattle and other Bovini species as the source populations. Our test using *B. javanicus* and Indian indicine cattle as the sources produced a Z-score of −26.97 (Supplementary Table 18), which was a highly significant value. *D* statistic tests were applied following the tree topology (Buffalo, Chinese indicine cattle; *B. javanicus*, other Bovini species), and the results suggested that Chinese indicine cattle was possibly introgressed from *B. javanicus*, producing a highly significant Z-score of −51.54 (Supplementary Table 19). Treemix also confirmed the gene flow from *B. javanicus* to Chinese indicine cattle (Supplementary Fig. 14). The same method used for the analysis of introgressive hybridisation within Mongolian yak[7] was employed to further assess *B. javanicus* introgression into Chinese indicine genomes (Fig. 5a, b). We used the number of *B. javanicus*-specific SNPs per 5-kb interval as an indicator to identify local introgressions (Supplementary Fig. 15). The SNP data were used to confirm the *B. javanicus* introgression into Chinese indicine cattle (Fig. 5d, e). Phylogenetic analyses of 12 haplotypes representing *B. javanicus*, Chinese indicine cattle, and 8 other Bovini species clustered Chinese indicine cattle (three individuals) with *B. javanicus*, thus confirming the *B. javanicus* introgression into the Chinese indicine genomes (Fig. 5g, h). We used whole-genome resequencing data of 17 Chinese indicine genomes for the systematic analysis of *B. javanicus* introgression into Chinese indicine populations. The proportion of the genome inferred to be of *B. javanicus* ancestry ranged between 2.38 and 3.84% (mean ± standard error (SE) = 2.93 ± 0.11%) per genome (Supplementary Table 20). We selected all non-coding regions in the

**Fig. 4** Relationship of ancient Chinese cattle to present-day cattle and genetic diversity of the five "core" cattle groups. **a** Neighbour-joining tree of the relationship between the ancient Chinese cattle and five "core" groups. The scale bar represents the pairwise distances between different individuals. **b** Shared drift with ancient Shimao cattle in 42 modern-day cattle populations using the outgroup-*f*3 statistics (Indicine cattle; Ancient, *X*). The indicine cattle include three individuals of Hariana, Sahiwal and Tharparkar cattle. **c** Genome-wide distribution of nucleotide diversity of five "core" cattle groups in 50-kb non-overlapping windows. The horizontal line inside the box corresponds to the median of this distribution, the bottom and top of the box are the first and third quartiles. Data points outside the whiskers can be considered as outliers. **d** Venn diagram showing unique and shared SNPs among the five "core" cattle groups

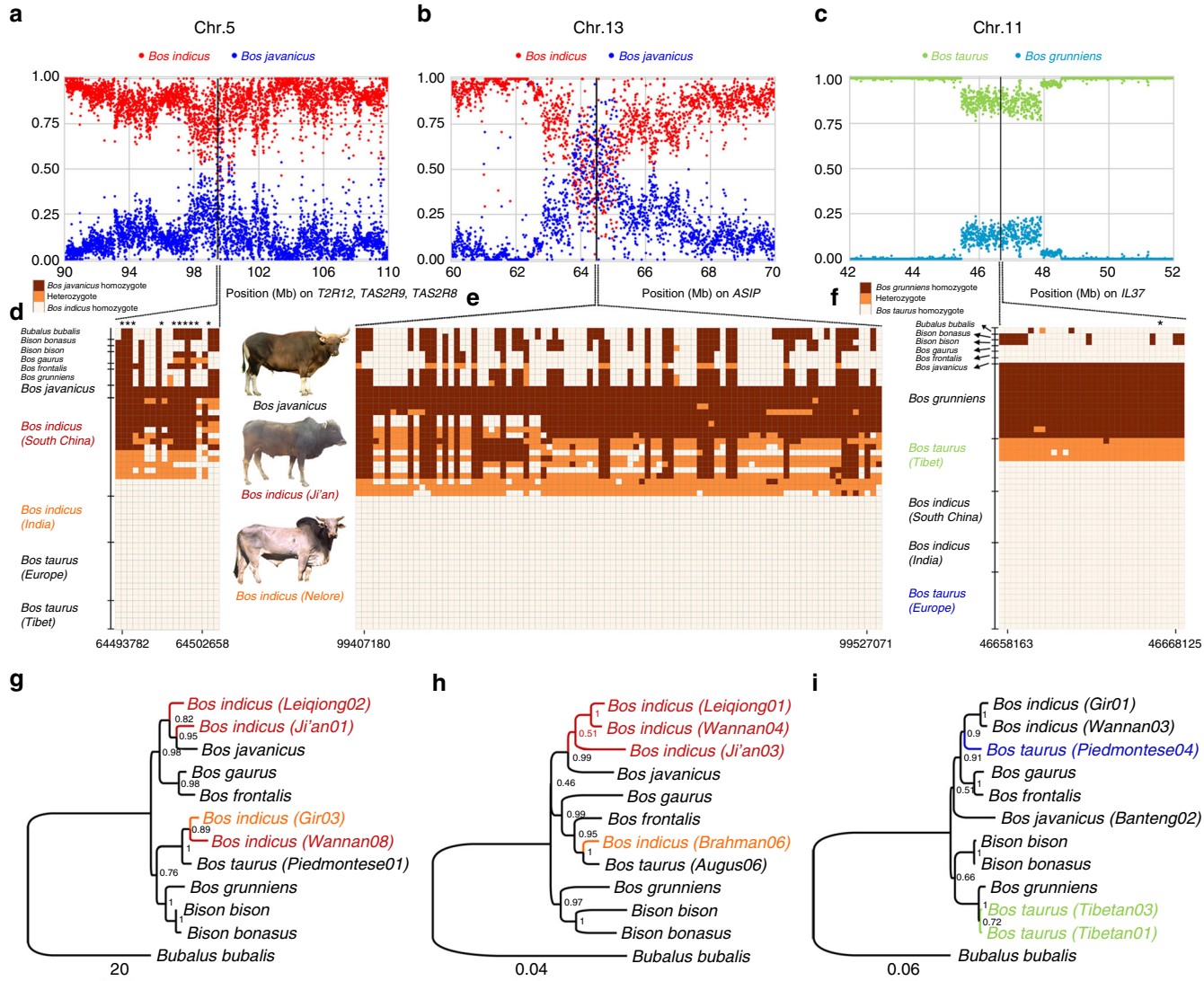

**Fig. 5** Phylogenetic analyses of SNP data confirm the *Bos javanicus* introgression into Chinese indicine cattle or the yak introgression into Tibetan taurine cattle. **a**, **b** *Bos javanicus* introgression plots constructed based on whole-genome sequencing data from two representative segments on chromosomes 5 and 13. The relative frequencies of *Bos indicus*-specific alleles are in red dots while *Bos javanicus*-specific alleles in blue dots. *Bos javanicus*-specific alternate alleles were identified as those that appeared in *Bos javanicus* genomes and were absent from taurine and Indian indicine cattle genomes. **c** Yak introgression plots constructed based on whole-genome sequencing data from a reprehensive segment on chromosome 11. The yak-specific alleles were identified using the same method. The relative frequencies of *Bos taurus*-specific are in green dots while *Bos grunniens*-specific are in sky blue. Vertical mouldings in **d**–**f** represent the SNP regions on chromosomes 5, 13 and 11 used for neighbour-joining (NJ) phylogeny analysis. The patterns of high-frequency SNPs at the *T2R12*, *TAS2R9* and *TAS2R6* regions are shown in **d**, at the *ASIP* region in **e** and at the *IL-37* region in **f**. The star (\*) denotes non-synonymous SNPs. Nucleotides with brown box in **d** and **e** represent *Bos javanicus*-specific homozygote genotypes, and with orange box represent heterozygote genotypes. Nucleotides with brown box in **f** represent *Bos grunniens*-specific homozygote genotypes. NJ phylogeny of 12 haplotypes supporting the introgression of banteng into Chinese indicine cattle (**g, h**) and the introgression of yak into Tibetan taurine cattle (**i**). The reliability of the tree branches (shown at the nodes) is tested with 1000 bootstrap replicates

segments introgressed in 15 or more Chinese indicine haploids to construct a NJ tree and date the introgression (Supplementary Note 5 and Supplementary Fig. 16). The estimated introgression time was ~2.9 thousand years ago (kya) (~1.4–3.8 kya, 95% confidence interval (CI)) assuming a mutation rate of $1.26 \times 10^{-8}$ per generation and a generation interval of 6 years[22,23]. To identify phenotypes that have undergone positive selection, we next exploited the gene content of 1852 introgressed intervals that were shared by at least two haplotypes in the Chinese indicine cattle (Supplementary Table 21). Introgressed genes were then analysed for their Gene Ontology (GO) categories and Kyoto

Encyclopaedia of Genes and Genomes (KEGG) pathways. The GO analysis identified a significant over-representation of genes involved in biological processes that contributed to natural killer cell activation involved in immune response (GO:0002323), the G-protein coupled receptor signalling pathway (GO:0007186) and sensory perception of smell (GO:0007608) (Bonferroni-corrected $P < 0.01$; Table 1 and Supplementary Table 22). The KEGG pathway analysis identified a major enrichment of genes involved in sensory perception (bta04740, olfactory transduction; bta04742, taste transduction) and immunity (bta05322, systemic lupus erythaematosus) (Bonferroni-corrected $P < 0.01$; Table 1

**Table 1 Summary of the results from the enrichment analysis of genes introgressed from banteng into Chinese indicine cattle or from yak into Tibetan taurine cattle**

| Category[a] | ID | Enriched term[b] | Gene count | Corrected P value[c] | Enrichment |
|---|---|---|---|---|---|
| Enrichment analysis of genes introgressed from banteng into Chinese indicine cattle | | | | | |
| KEGG | bta04740 | Olfactory transduction | 269 | $1.2 \times 10^{-5}$ | 1.3 |
| KEGG | bta04742 | Taste transduction | 22 | $3.3 \times 10^{-4}$ | 3.0 |
| KEGG | bta05322 | Systemic lupus erythaematosus | 60 | $1.2 \times 10^{-3}$ | 1.8 |
| GO term | GO:0002323 | Natural killer cell activation involved in immune response | 30 | $1.3 \times 10^{-10}$ | 4.2 |
| GO term | GO:0007186 | G-protein-coupled receptor signalling pathway | 224 | $6.5 \times 10^{-10}$ | 1.6 |
| GO term | GO:0007608 | Sensory perception of smell | 69 | $1.1 \times 10^{-3}$ | 1.8 |
| Enrichment analysis of genes introgressed from yak into Tibetan taurine cattle | | | | | |
| KEGG | bta05332 | Graft-vs.-host disease | 16 | $2.8 \times 10^{-8}$ | 8.7 |
| GO term | GO:0007608 | Sensory perception of smell | 33 | $1.1 \times 10^{-7}$ | 3.9 |
| GO term | GO:1903352 | L-ornithine transmembrane transport | 9 | $2.5 \times 10^{-6}$ | 19.3 |
| GO term | GO:0002504 | Antigen processing and presentation of peptide or polysaccharide antigen via MHC class II | 9 | $5.1 \times 10^{-4}$ | 11.8 |

[a]The GO and KEGG analyses performed with DAVID 6.7 use different lists of genes present in chromosomal regions detected as introgressed from banteng into Chinese indicine cattle or from yak into Tibetan taurine cattle according to RFMix analyses
[b]One significantly enriched term is chosen from each group of significantly enriched intercorrelated terms
[c]P values are Bonferroni-corrected P values $\leq 10^{-2}$

and Supplementary Table 22). One introgressed region included the well-known coat colour gene *ASIP*, which has been implicated as a strong candidate gene that controls coat colour patterns in livestock and may partly explain the colour patterns of Chinese indicine cattle (Fig. 5b, e). No non-synonymous SNP was found within the region, suggesting that the potential target for selection might be regulatory mutations. Next, we detected 326 missense mutations in introgressed regions that segregated in Chinese indicine cattle. Of these, 10 non-synonymous substitutions were present within *T2R12*, *TAS2R9* and *TAS2R6* genes (Fig. 5d). These genes are functionally relevant to bitter taste in humans and giant pandas[24,25] and may have a similar role in Chinese indicine cattle. We also identified several introgressed genes that favoured local adaptation of Chinese indicine cattle to tropical climate environments (Supplementary Fig. 19 and Supplementary Table 23). For example, we found several heat-shock protein (HSP) genes, including *HSPA1A*, *HSPB8*, *HSPA8*, *HSPA4*, *HSPB2* and *HSF2*, which are involved in the key cellular defence mechanisms during the exposure to hot environments[26]. We detected several genes related to hair cell differentiation and blood circulation, including *ATOH*, *GNA14*, *VPS13* and *KIF2B*, which also play an important role in the temperature adaptation of Chinese pigs[27].

**Yak introgression into Tibetan taurine cattle**. The presence of yak mtDNA in Tibetan taurine cattle and the results of the Treemix, *f3* statistics, and *D* statistics analyses suggested similar yak introgressions (Fig. 3b, Supplementary Figs. 14 and 18, Supplementary Tables 24 and 25). The genome proportion of nine Tibetan taurine cattle inferred to be of yak ancestry ranged between 0.05 and 2.94% (mean ± standard error (SE) = 1.22 ± 0.18%) per genome (Supplementary Table 26). The estimated introgression time was ~ 1.9 kya (1.4 to 2.4 kya, 95% CI). We exploited the gene content of 2395 introgressed intervals that were shared by at least two haplotypes in Tibetan taurine cattle (Supplementary Table 27). The functional categories that were enriched for significantly introgressed genes mainly included sensory perception of smell (GO:0007608), L-ornithine transmembrane transport (GO:1903352), and antigen processing and presentation of peptide or polysaccharide antigens via major histocompatibility complex class II (GO:0002504) (Bonferroni-

corrected *P* < 0.01; Table 1). The KEGG annotation showed that the largest groups of introgressed genes were involved in disease resistance (bta05323, graft-vs.-host disease) (Bonferroni-corrected *P* < 0.01; Table 1 and Supplementary Table 22). A total of 392 missense mutations were detected in introgressed regions. Among these genes, we highlighted *IL-37*, a fundamental inhibitor of innate immunity[28], which may exert anti-inflammatory effect during intestinal inflammation[29] (Fig. 5c, f, i). In addition, we retrieved several hypoxia genes in introgressed regions that may help Tibetan cattle adapt to hypoxic environment (Supplementary Fig. 20 and Supplementary Table 28). These included candidate genes of *COPS5*, *IL1A*, *IL1B*, *MMP3* and *EGLN1* for the hypoxia-inducible factor pathways that have been repeatedly identified among targets for selection to high-altitude adaptations in Andeans, Tibetans and yak[30–32]. We also observed two genes, *RYR2* and *SDHD*, to be involved in mediating calcium homeostasis that regulates the response to hypoxia[33].

**Demographic history of East Asian cattle**. The multiple sequentially Markovian coalescent (MSMC) approach[34] with a generation time of $g = 6$ and a mutation rate per generation of $\mu_g = 1.26 \times 10^{-8}$ [22,23] was used to reconstruct the population history of the five "core" cattle groups (Supplementary Note 6). We applied this method to all cattle groups each with two deep-coverage (>15×) individuals. To evaluate the impact of introgression on the estimates of effective population size ($N_e$) and divergence time, we repeated the MSMC analysis using the same data but excluding the introgressed regions from banteng to two Chinese indicine cattle (~3.5 %) or from yak to two Tibetan taurine cattle (~1.3 %) (Supplementary Tables 20 and 26), respectively. The results showed that the limited introgression did not notably change the estimates of divergence time and $N_e$ (Supplementary Fig. 21).

For both taurine and indicine cattle, a common dramatic decline in $N_e$ was detected at 20–30 kya, which likely reflects the major climatic change at the end of the last glacial maximum[35] (Fig. 6a), predating cattle domestication. We also observed a decline of $N_e$ during 7–9 kya consistent with the onset of domestication (Fig. 6a).

We then used MSMC to calculate the divergence times among the five "core" cattle groups defined in this study. We calculated a

decrease in the cross-coalescence rate between *B. taurus* and *B. indicus* to 0.5 at approximately 201 to 213 kya (Fig. 6b). Our estimated time overlapped with the molecular divergence of taurine and indicine lineages, which was dated between 117 and 332 kya[21,36]. The relative cross-coalescence analysis suggested a decline to 0.5 between East Asian taurine (Tibetan) and Eurasian taurine (Gelbvieh) or European taurine (Hereford) cattle at ~ 6.6 kya (0.25 to 0.75 range = 4.5 to 13.1 kya for Eurasian taurine

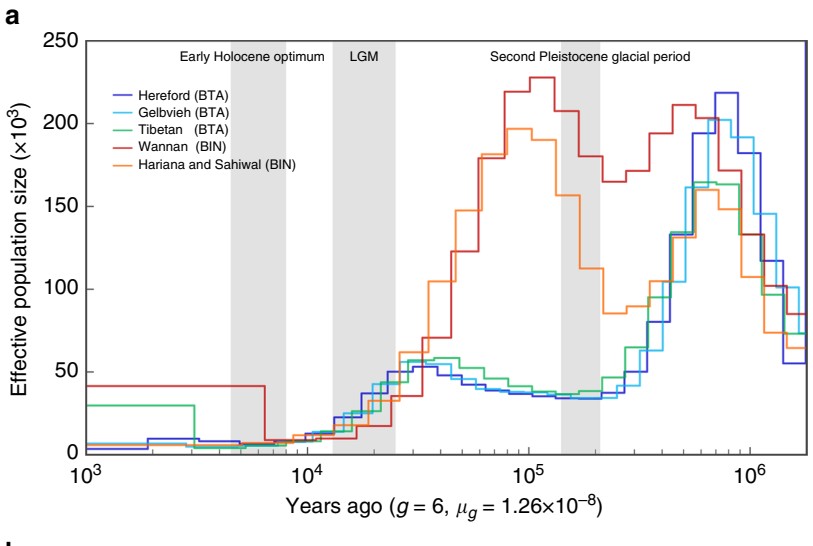

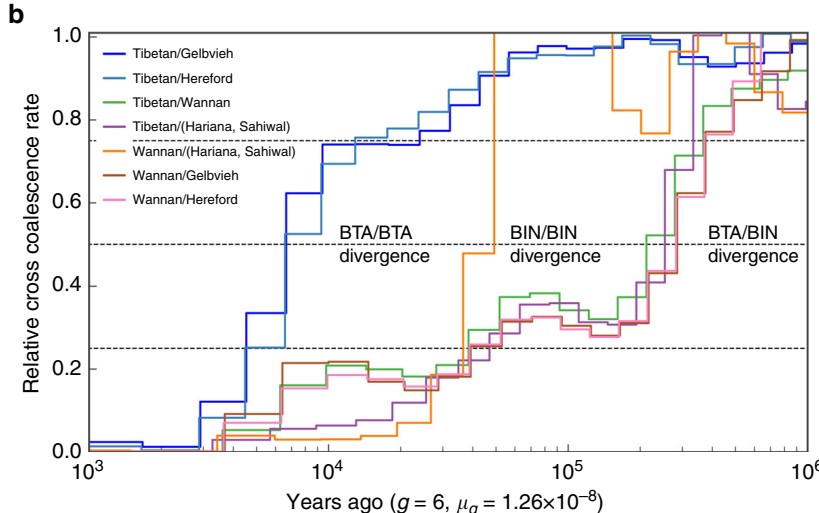

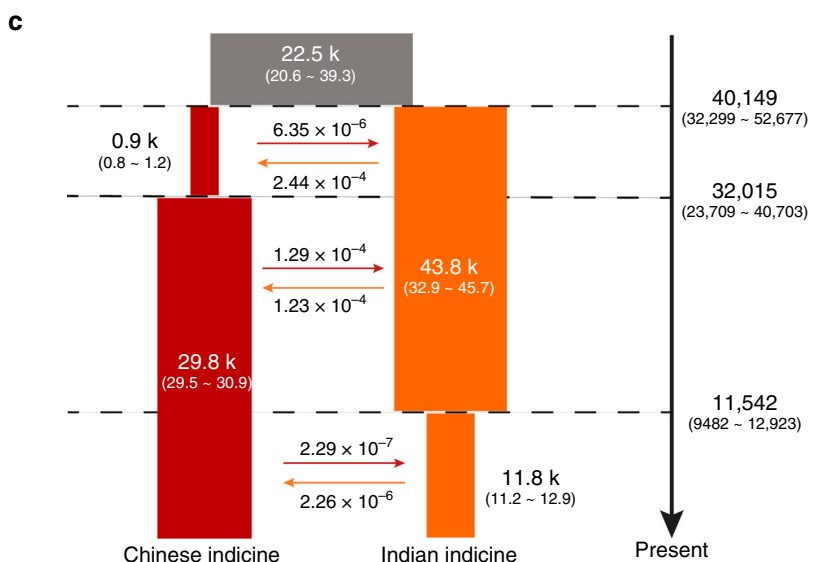

cattle; 0.25 to 0.75 range = 4.5 to 12.9 kya for European taurine cattle). However, we observed an earlier and clearly pre-domestication split ~ 49.6 kya (0.25 to 0.75 range = 36.6 to 49.6 kya) between Chinese indicine (Wannan) and Indian indicine cattle (Hariana and Sahiwal). The simulation of diffusion approximations for demographic inference (∂a∂i)[37] showed that the Chinese indicine and Indian indicine populations diverged ~ 40.1 kya (~ 30.2 to 52.6 kya, 95% CI; Fig. 6c and Supplementary Fig. 22), which gave rise to today's pattern of two genetically distinct indicine backgrounds in Asia.

## Discussion

Previous population analyses based on nuclear markers have confirmed the genetic ancestry of European, African, Indian and American cattle[4–7]. However, the histories of East Asian cattle populations based on genome level analysis have been poorly examined. The following five major ancestries (and paternal lineages) were consistently observed in our study. In addition to European taurine (Y1) and Indian indicine (Y3b), three types of ancestry, including Eurasian taurine (Y2a) and two other distinct ancestries (East Asian taurine (Y2a) and Chinese indicine (Y3)), were observed in East Asian cattle (Fig. 2c). Phylogenetic analyses suggested that present-day East Asian domesticated cattle mainly originated from three bull populations. The north-western populations shared the Y2a sub-haplogroup with the Central-South Europe populations, thus representing Eurasian taurine ancestry, whereas the sub-haplogroup Y2b was predominant in cattle from Northeast Asia and Tibetan Plateau, thus representing East Asian taurine ancestry. Cattle in South China are primarily of *B. indicus* ancestry. They belong to the sub-haplogroup Y3a (Fig. 2b, c) and also carry a maternal I1a lineage. Our results show that paternal lineages have a clear phylogeographical structure, which concords with autosomal ancestral components.

Archaeological evidence indicates that two major domestication events of cattle occurred 7000–10,000 YBP in the Fertile Crescent and Indus Valley[3,38]. Under this hypothesis, all present-day East Asian cattle may be descended from the domesticated cattle from these two areas[6,8,36,38]. In this study, we infer that two distinct *B. taurus* ancestries influenced East Asian cattle. The ancestral component labelled Eurasian taurine and the Y2a sub-haplogroup were observed at the highest frequency in Northwest China and Central-South European cattle, whereas the Tibetan and Northeast Asian breeds shared the more narrowly distributed East Asian taurine ancestry component and the Y2b sub-haplogroup. This clear phylogeographical structure and the markedly strong affinity of the East Asian cattle to an ancient Chinese sample suggest that *B. taurus* migration did not occur through a single event. The earlier strand of ancestry was introduced at least ~3.9 kya or several millennia before[5,13,39,40]. Later on, the East Asian taurine cattle might become widespread all over northern China and margins of the Tibetan Plateau through broader expansion of millet agriculture around 3.6 kya[40,41]. The earlier strand may have migrated from Northeast China to Japan via the Korean peninsula at quite a late stage in the second

century AD[42]. The Eurasian strand possibly was introduced and spread later by the expansion of early pastoralism, leading to west-to-east immigrations, which may explain the similar component of Southern European and Mongolian cattle[43]. In addition, the recent introgression of Mongolians into huge and extensive areas of China may have also led to gene flow across eastern Eurasia and partial replacement of East Asian taurine ancestry (and the Y2b lineage) in northern China. Today, the narrow distribution of pure East Asian taurine cattle in Tibetan Plateau and Northeast Asia might be maintained by geographical barriers.

The *B. javanicus* species historically ranged throughout the southeast mainland and southern China[44]. We observed that Chinese indicine cattle inherited a ~2.93 % genome component from *B. javanicus* ancestry at least 2.9 kya, which coincides with the earliest evidence for indicine cattle in China 3 kya[14,15]. The initial period of introgression is most likely due to the eastward migration of indicine cattle to China[45]. However, the impact of other species on Chinese indicine cattle cannot be completely excluded[46]. Mitogenomic analyses showed narrow diversity within Chinese indicine cattle, suggesting a strong matrilineal founder effect after their arrival to South China.

Two types of *B. indicus* ancestry were clearly supported by autosomal and Y-chromosome evidence and they diverged 36.6–49.6 kya, indicating that Chinese and Indian indicine cattle might be descendants from divergent wild populations. Given our combined results, we suggest the following hypothesis: two genetically differentiated wild *B. indicus* populations may have contributed to indicine ancestry or Chinese indicine cattle had input from a separate strain of Indian aurochs through eastward migration. Nevertheless, given the complexity of the evolutionary history of bovine species, hybridisation introgression and incomplete nature of the archaeological records, this scenario remains hypothetical. With this in mind, further geographically informed whole-genome analysis of bovine species, coupled with ancient DNA study, will reveal a more clear landscape of the complex cattle domestic histories in East Asia.

Recently, several studies have supported the idea that introgression plays a potentially important role in adaptation in humans and yak[16,27,47,48]. *B. javanicus* are well adapted to the tropical climate environment, food and local pathogens. We show here that several introgressed immune, sensory and heat adaptation genes into Chinese indicine cattle may have acquired advantageous alleles via admixture with *B. javanicus*. Similar introgression was also found from yak into Tibetan taurine cattle. These introgressed segments are enriched in genes involved in disease resistance and response to hypoxia, which probably have contributed to the adaptation of Tibetan taurine cattle to extreme environmental conditions of the Tibetan Plateau. In addition, a recent study confirmed that British aurochs may have changed the nuclear composition of local cattle[49]. Ancient aurochs were present in northern China around 3850–10,660 YBP[50,51], implying that opportunities occurred for interbreeding and experimentation via the management of aurochs. Regardless of

**Fig. 6** Coalescence-based inference of demographic history of cattle using MSMC and ∂a∂i. **a** Population size history inference of *Bos taurus* (BTA) and *Bos indicus* (BIN) lineages based on four haplotypes each from high-coverage European taurine (Hereford), Eurasian taurine (Gelbvieh), East Asian taurine (Tibetan), Chinese indicine (Wannan) and Indian indicine (Hariana and Sahiwal) individuals. The large grey-shaded boxes illustrate the Early Holocene Optimum, the last glacial maximum (LGM), and the second Pleistocene Glacial Period. **b** Inferred relative cross-coalescence rates between pairs of populations over time based on four haplotypes each from Hereford, Gelbvieh, Tibetan, Wannan and Indian breeds (Hariana and Sahiwal). The x-axis shows time and the y-axis a measure of similarity for each pair of compared populations. **c** ∂a∂i result showing the divergence time of Chinese indicine and Indian indicine cattle. The ancestral population is in grey, Chinese indicine cattle is in red and Indian indicine cattle is in orange. The width shows the relative effective population size. The figures at the arrows indicate the average number of migrants per generation between Chinese indicine and Indian indicine cattle

whether the introgressions occurred naturally or by human management, introgressive hybridisation increased the variability of East Asian cattle and generated a valuable genetic resource for modern cattle breeding, and these findings are consistent with the beneficial role of introgressive hybridisation in the improvement of yak breeding[16].

In conclusion, our population genomic analyses of East Asian native cattle provide new insights into their origin, historical migrations and introgression events. The East Asian cattle have three distinct ancestral lineages: East Asian taurine; Eurasian taurine; and Chinese indicine. Notably, our study supports the hypothesis that adaptive introgressions from other bovine species to cattle represent an important source of genetic variation in natural populations and may contribute to adaptations in domestication process.

## Methods

**Genome sequencing.** We sampled a total of 114 cattle, including 111 Chinese domestic cattle (1–9 individuals for each of the 22 geographically diverse populations) and 3 indicine cattle in India (Fig. 1a and Supplementary Tables 1 and 2). DNA was extracted from the ear tissues or blood of each individual. Paired-end libraries with insert size of 500 bp were constructed for each individual and sequenced using the HiSeq 2000 platform (Illumina) (Supplementary Note 1). In addition, we downloaded the genome data of 146 individuals across the world from the NCBI database, including 83 European taurine cattle, 26 Northeast Asian taurine cattle, 8 Iranian cattle, 17 American indicine cattle and 12 African cattle (Supplementary Tables 1 and 2). All experimental procedures were performed in accordance with the Regulations for the Administration of Affairs Concerning Experimental Animals approved by the State Council of People's Republic of China. The study was approved by Institutional Animal Care and Use Committee of Northwest A&F University (Permit number: NWAFAC1019).

**Read mapping and SNP calling.** First, all cleaned reads were mapped to the cattle reference assembly Btau_5.0.1 (GCF_000003205.7) using BWA-MEM (0.7.13-r1126) with default parameters. The average mapping rate of the reads sequenced in this study was 98.84%, and the sequencing coverage was approximately 12.72× (ranging from 3.92 to 36.85) per individual. Duplicate reads were removed using Picard Tools. Then, the Genome Analysis Toolkit (GATK, version 3.6-0-g89b7209) was used to detect SNPs. The following criteria were applied to all SNPs: (1) mean sequencing depth (for all individuals) >1/3× and <3×; (2) variant confidence/quality by depth > 2; (3) RMS mapping quality (MQ) > 40.0; (4) Phred-scaled $P$ value using Fisher's exact test to detect strand bias < 60; (5) $Z$-score from the Wilcoxon rank sum test of Alt vs. Ref read MQs (MQRankSum) > −12.5; and (6) $Z$-score from the Wilcoxon rank sum test of Alt vs. Ref read position bias (ReadPosRankSum) > −8 (Supplementary Note 1 and Supplementary Table 3).

**Population genetic structure and admixture.** The PCA of the SNPs was performed using the smartpca programme in EIGENSOFT v5.0[52]. The Tracy-Widom test was used to determine the significance level of the eigenvectors. ADMIXTURE version 1.3.0 was used to quantify the genome-wide admixtures among modern cattle populations[53]. ADMIXTURE was run for each possible group number ($K = 2$ to 8) with 200 bootstrap replicates. For autosomal genome data, a NJ tree was constructed with PLINK (version 1.9) using the matrix of pairwise genetic distances. We also inferred a population-level phylogeny using the ML approach implemented in TreeMix[54]. To distinguish the groups that may have hybridised with Chinese indicine and Tibetan taurine cattle, ADMIXTOOLS (version 4.1) was used to perform $f3$ and $D$ statistical analyses[55,56]. ADMIXTOOLS were used to detect relationship between the ancient cattle and modern cattle groups (Supplementary Note 2).

**Paternal analysis.** We selected the X-degenerate region that consists of single-copy genes within the male-specific part of the Btau_5.0.1 Y-chromosome reference sequence (GCF_000003205.7)[18]. After removing heterozygous sites and sites with missing genotypes in 10% of the sampled individuals, we extracted 745 SNPs. FASTA-formatted sequence files were used to generate haplogroup trees. Sequence alignments were built using CLUSTALW2 (http://www.clustal.org/), and a maximum parsimony phylogenetic tree was created using PHYLIP (http://evolution.gs.washington.edu/phylip.html). We also used BEAGLE (version 4.1) to impute missing alleles and checked them according to the tree structure[57]. Additional phylogenetic trees were then inferred using both ML and Bayesian methods.

**Whole-mitochondrial genome phylogeny.** We assembled 147 complete mitochondrial genome sequences from the whole-genome resequencing data and aligned them to a previously published collection of 24 mitochondrial genomes of the genus *Bos*, which were selected to encompass the whole spectrum of the known

mitochondrial diversity reported thus far within *B. taurus*, *B. indicus*, *Bos gruniens* and *Bison bision* species. The best substitution models were determined using Modelgenerator version 0.85[58] for the full mitochondrial alignments. ML phylogenetic inferences were performed in PhyML3.0[59] using the best substitution models. Bayesian analyses were also performed on the concatenated partitions in BEAST 1.8.0[60] (Supplementary Note 3).

**Ancient genome data processing.** DNA was successfully extracted from eight fossil cattle specimens excavated in the late Neolithic Shimao site in northern China (3975–3835 cal BP). The humerus bone from specimen Shimao05 was sampled for radiocarbon dating analysis. Ancient DNA extracts were built into Illumina libraries and sequenced on the Illumina® HiSeq X Ten platform. Reads were mapped to the bovine reference genome Btau_5.0.1 (GCF_000003205.7).

**Introgression analysis.** To detect introgressions from *B. javanicus* into Chinese indicine cattle, we first identified *B. javanicus*-specific alternate alleles that appeared in *B. javanicus* but were absent from other domesticated taurine and Indian indicine cattle. Then, 5-kb sliding windows were used to calculate the mean frequency of *B. javanicus* and indicine alleles in the Chinese indicine cattle (Supplementary Note 5 and Supplementary table 29). To identify introgressed intervals in Chinese indicine genomes, we applied a robust forward-backward algorithm for all autosomes of Chinese indicine genomes using the software RFMix[61]. Five other genetic groups were selected as reference panels, including European taurine, Eurasian taurine, Eastern Asian taurine and Indian indicine cattle as well as banteng. The software bpp3.3a was used to estimate the introgression time under the multispecies coalescence model on a fixed species phylogeny with the following Gamma priors: $\theta \sim G$ (2700), $\tau \sim G$ (20, 4,000,000)[62]. The same method was used for detecting the introgression from yak into Tibetan taurine cattle (Supplementary Note 5).

**Annotation of the gene content of the introgressed segments.** To provide a first overview of the over-represented groups of genes, we performed GO and KEGG analyses with DAVID 6.7 (http://david.abcc.ncifcrf.gov/) using lists of genes located in the chromosomal regions detected as introgressed from banteng into Chinese indicine cattle or from yak into Tibetan taurine cattle by RFMix analyses. The $P$ values are Bonferroni-corrected $P$ values. Only pathways or annotations with $P < 10^{-2}$ were retained.

**Estimates of the effective population size and divergence time.** The MSMC method was used to model the history of the five genetic groups and infer the historical changes in their effective population sizes and population separations. We applied this method to all groups with four haplotypes per group (Supplementary Note 6). For each individual, we identified SNPs in the autosomes using GATK, and sites with extremely low or extremely high coverage were excluded. All sites were phased using BEAGLE version 4.1[57]. We defined the estimated divergence time between a pair of populations as the first time point at which the cross-coalescence rate was at or above 0.5. For the range of divergence times, we used the first time point at which the cross-coalescence rate was at or above 0.25 and 0.75. The timescale in generation time at $g = 6$ and a mutation rate per generation at $\mu_g = 1.26 \times 10^{-8}$ were used in the modelling.

We used $\partial a \partial i$ approach to simulate the divergent time of Chinese indicine and Indian indicine backgrouds[37] (Supplementary Note 6). The simulation results of the model are shown in Supplementary Table 30.

**Data availability.** Raw FASTQ sequences have been deposited to NCBI Short Read Archive under the BioProject accession number PRJNA379859. The data that support the findings of this study are available from the corresponding authors upon reasonable request.

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

## Acknowledgements

The project was supported by the programmes of the National Beef Cattle and Yak Industrial Technology System (CARS-37) to C.L., National Thousand Youth Talents Plan to Y. Jiang, the National Natural Science Foundation of China (31501918) to X.Y., Natural Science Foundation of Qinghai Province of China (2017-ZJ-906) to Z.M. and Yunnan Modern Agriculture Beef Cattle Industry Technology System (2017KJTX0015) to W.D. We also thank members of the NextGen project for sharing their data. We thank Alessandro Achilli, Ben J Hayes, Stephen Moore and Xihong Wang for helpful discussion.

## Author contributions

Y. Jiang and C.L. designed and supervised the project. N.C. and Y.C. performed the majority of analysis with contributions from Q.C., R.L., K.W., S. Huang and Z. Zheng. Y.H., Z.M., Y.M., R.D., Z. Zhang, L.X., Y. Jia, S.L., X.Y., W.D., X.L., H.C., D.G.B. and C.L. prepared the modern samples. S. Hu and Z.S. prepared the ancient samples. N.C., Y. Jiang and C.L. wrote the manuscript with imput from all authors, whereas R.L., H.Z., W.S., X.Z., J.H. and D.G.B. revised the manuscript.

## Additional information

**Competing interests:** The authors declare no competing interests.

Ningbo Chen[1], Yudong Cai[1], Qiuming Chen[1], Ran Li[1], Kun Wang[2], Yongzhen Huang[1], Songmei Hu[3], Shisheng Huang[1], Hucai Zhang[4], Zhuqing Zheng[1], Weining Song[5], Zhijie Ma[1,6], Yun Ma[7], Ruihua Dang[1], Zijing Zhang[8], Lei Xu[9], Yutang Jia[9], Shanzhai Liu[10], Xiangpeng Yue[11], Weidong Deng[12], Xiaoming Zhang[13], Zhouyong Sun[3], Xianyong Lan[1], Jianlin Han[14,15], Hong Chen[1], Daniel G Bradley[16], Yu Jiang[1] & Chuzhao Lei[1]

[1]Key Laboratory of Animal Genetics, Breeding and Reproduction of Shaanxi Province, College of Animal Science and Technology, Northwest A&F University, Yangling 712100, China. [2]Center for Ecological and Environmental Sciences, Northwestern Polytechnical University, Xi'an 710129, China. [3]Shaanxi Provincial Institute of Archaeology, Xi'an 710054, China. [4]Key Laboratory of Plateau Lake Ecology and Environment Change, Yunnan University, Kunming 650504, China. [5]State Key Laboratory of Crop Stress Biology in Arid Areas, Yangling Branch of China Wheat Improvement Center, College of Agronomy, Northwest A&F University, Yangling 712100, China. [6]Academy of Animal Science and Veterinary Medicine, Qinghai University, Xining 810016, China. [7]Agricultural College, Ningxia University, Yinchuan 750021, China. [8]Institute of Animal Science and Veterinary Medicine, Henan Academy of Agriculture Science, Zhengzhou 450002, China. [9]Institute of Animal Science and Veterinary Medicine, Anhui Academy of Agriculture Science, Hefei 230001, China. [10]Bozhou Comprehensive Experimental Station, National Beef Cattle and Yak Industrial Technology System, Bozhou 236000, China. [11]State Key Laboratory of Grassland Agroecosystems, College of Pastoral Agriculture Science and Technology, Lanzhou University, Lanzhou 730000, China. [12]Faculty of Animal Science and Technology, Yunnan Agricultural University, Kunming 650201, China. [13]State Key Laboratory of Genetic Resources and Evolution, Kunming Institute of Zoology, Chinese Academy of Sciences, Kunming 650223, China. [14]CAAS-ILRI Joint Laboratory on Livestock and Forage Genetic Resources, Institute of Animal Science, Chinese Academy of Agriculture Sciences (CAAS), Beijing 100193, China. [15]International Livestock Research Institute (ILRI), Nairobi 00100, Kenya. [16]Smurfit Institute of Genetics, Trinity College Dublin, D02 DK07 Dublin, Ireland. These authors contributed equally: Ningbo Chen, Yudong Cai, Qiuming Chen, Ran Li, Kun Wang, Yongzhen Huang.

