## [Peer Review File · Nature Communications]

Reviewers' comments:

Reviewer #1 (Remarks to the Author):

This is a very interesting study that brings new information to understand the complex history of cattle domestication. Even if it focusses on East Asian breeds, it gives a quite comprehensive view of the events that occurred in this region (diversity of wild stocks contributing to domestication, migrations, introgressive adaptation) and supports the importance of such mechanisms in shaping modern cattle. Excepting a few restrictions (see below) the study is based on a reliable sampling with regards to the question addressed. The results appear to be robust as they are generally supported by several analyses converging in supporting the same conclusion. This study should be of interest for a wide audience.

1. Main concerns.

My main concern is about the introgression from *Bos javanicus* in Chinese indicine cattle. First, I do not challenge the occurrence of such hybridization that has already been reported (e.g., Hartati et al. 2015, doi.org/10.1186/s12863-015-0229-5), but I wonder how the sampling might affect the results obtained. Only two bantengs were used as representative of the species, and they were from a zoo. First this is a really low sample size that might not represent well the species even more if the 2 individuals are closely related. Information on relatedness and inbreeding should be given. Moreover, we know that hybridization between close species might occur in captivity, and this could impact the result obtained. The authors should provide more information on these samples/genomes and discuss how this can impact the results. The same problem is for *Bos gaurus*, but with less impact as no introgression from this species occurs.

A second point is about the adaptive aspect of the introgression. In the title and abstract, the authors refer to 'adaptive' introgressions. However this is speculative, as the genes found in introgressed regions are not even related to environmental changes. To confirm such adaptive process for at least some genes, the authors should link the presence of banteng-specific alleles in cattle to the occurrence of given environmental conditions. Otherwise, they cannot claim that this is an adaptive process.

2. Other concerns.

- A global concern is about Figure and Table numbering. In both the main text and supplementary material the Fig and Tab numbers should be checked and modified when necessary. It is sometimes difficult to find the good information as the item number do not correspond : e.g. suppl. Table 5 refers to the PCA not to SNPs (line 135), Suppl. Table 8 do not exist, etc. ; Suppl. Table 7 (line 172) should correspond to Suppl Table 10; also Figure 3 is called before Figure 2 in the main text. Figures would be more easily interpreted by the reader with a few more information, such as for example the indicine/taurine origin of the breed in Fig. 1, Fig. 6a, Suppl. Figures 1-5.

- More details should be given for some methods :

suppl. note 1 - It is not clear whether the genotype data imported from public databases were used for variant discovery or not. The set of genomes used for variant discovery should be stated (as it has an impact on the polymorphism revealed). What kind of information was retrieved from the databases ?

suppl. note 2 - it is not clear if all fourfold-degenerate sites were chosen or if it is a subsample ; the distance used for building the NJ tree should be given ; information on how was build the ML tree should be provided.

- The interpretation linking the bottleneck inferred from MSMSC analyses at 10 kya to domestication (l 273) is not straightforward as the low N_e at this time follow a continuous demographic decline that began at least 20 kya before.

3. Minor points:

Main document

- line 50. Here 'aurochs' are actually Indian aurochs (ie indicine subspecies). Should be clearer if explicit I think.
- line 99. I would find 'analysis of the genetic structure' more convenient than 'population genetic...'
- line 204. remove 'remarkably'
- lines 293-295. The sentence is not clear, please rephrase.
- line 306. I think that 'areas' would be better adapted than 'sites'
- lines 324-328. The sentence is not clear, please rephrase.
- lines 346-347. Would be clearer if reminding here that the introgression in Tibetan taurine is from Yak.
- line 351. 'improve the variability'. Do you mean 'increase'?

Supplementary Material

- lines 45-49. Not clear. Please rephrase
- line 200. what do you mean by 'genetically pure' individuals ?
- lines 270-272 and 272-275. These two sentences are not really clear. Please rephrase
- line 336. 'individual' -> individual
- Suppl. Table 22. I wonder why many contiguous segments were considered independently and not as a single region (e.g., chr 1 6336595 -10025102, 10025102-10327999, 10327999 - 110722931 , etc.). Is it related to previous haplotype definition?

Reviewer #2 (Remarks to the Author):

Chen et al. analyzed whole-genome resequencing data of hundreds of cattle, including 8 ancient cattle from East Asia. They classify cattle into 5 groups. They see a split in Asian taurine cattle and a split in indicine cattle. They estimate that the split between Chinese indicine and Indian indicine occurred approximately 36 to 49 kya. They also note banteng and yak introgression into Asian cattle.

The authors claim that the histories of East Asian cattle are poorly understood. However, some of the events they describe have been previously reported. Banteng introgression into Chinese indicine has previously been reported by Decker et al. 2014 (Chinese Hainan and Luxi breeds).

I also suspect of the divergence date between Indian and Chinese indicine cattle. The fit between the dadi model and the observed data doesn't look great. The model has the weakest fit for variants that are at high frequency in the Chinese samples and moderate frequency in the Indicine samples. It also has a weak fit for rare variants in Indian samples that are at a moderate frequency in Chinese. The Wanna have the highest effective population size in the MCMC results. However, in the dadi model, they are estimated to be smaller than the Indian indicine Ne.

On page 15, lines 299 through 302, there is a discussion of male-mediated gene flow. If the females were stationary and the males were moving, wouldn't the mitochondria DNA reflect geography? However, what could be occurring is that females are being moved (likely from domestication centers) and introgression from yak, banteng, and possibly regional auroch are creating the differences in chromosome Y lineages? Perhaps this is what the authors intended to say and I didn't follow. Regardless, this section should be clarified.

On lines 317 to 320 of the Discussion, the population structure of Eurasian taurines is discussed. Could trade along the Silk Road influenced the relationships between Southern European (Italian, etc.) cattle and Mongolian and North-Central China cattle?

Additional comments:

Line 266. Why is mutation rate reported per year?

Line 285, misspelling of Indian.

Reviewer #3 (Remarks to the Author):

General Comments:

In general this is a good paper that in my opinion is well worth publishing in a journal such as Nature Communications subject to the revisions/clarifications described below. I enjoyed reading it. Though new methods are not developed, existing ones are applied appropriately, and a lot of new data is generated. The conclusions are sound and supported by the data, and interesting from the perspective of researchers trying to understand the process of domestication (the parts on introgression are particularly interesting).

My comments below are meant primarily to help improve the reading of the manuscript, especially for non-cattle experts such as myself, as sometimes this is not clear. The only two disappointing things from the study are a) the massive gap in sampling between Europe and China (there is nothing from Central Asia), which would help better contextualize the results, especially for the "Eurasian" component, and the very limited analysis of the ancient DNA, which to me is not fully exploited (how did some functional alleles look for example?). If there is Central Asian data is available, I would strongly urge the authors to include this.

Specific Comments:

Line 64: Sentence beginning: "Nuclear markers confirmed genetic ancestry"

Could do with a little more detail on what the patterns were in these previous studies (i.e. existing knowledge of cattle genetic structure. Similarly, a little explanation on the background/known differences between taurine (Taurus) and zebu (indicus) are needed for non-specialists

Line 74: "However, several ancient Chinese aurochs samples have yielded a highly divergent haplogroup, implying a minimal local matrilineal incorporation and complex histories of East Asian cattle."

Not sure what the link is here with the previous statement, and how the ancient data reflect complex histories.

Fig 1 needs more information for interpreting the results more easily. Which points are Taurus and which are Indices (I assume red and orange but had to guess). Also, why is yellow considered southwest china and red south china,

Line 95: "Among these genomes, we detected a total of 60.4 million putative autosomal SNPs" What is the importance of this, especially given the uneven coverage?

Line 102: "and the second component was driven by a split of East Asian cattle from other individuals"

Is this really the case? The major split appears to be between India/Pakistan/African versus everyone else. East Asian cattle show some structure but this isn't surprising given there are many more of them than elsewhere (sample number has an influence on PCA structuring due to increased covariance amongst samples).

Line 105: "Southwest China cattle exhibited intermediate ancestry" They seem closer to south china than India/Pakistan

Line 106: "A separation was also found between European and East Asian taurine cattle along the third component"

I do not see this, European cattle seem to have the same PC3 co-ordinates as NC and SW China.

Line 108: "The same population affinities were recovered in trees constructed by the neighbour-joining (NJ) method"

This structure is very hard to see as the branch lengths are so small. Consider removing to supp and making bigger.

Lines 107-129: Why is Fig 3 not presented as Fig 2 in terms of text order? In general I think this section needs a little bit of cleaning up and perhaps shortening. There is some repetition with regards to the grouping.

Line 136: "three common Y haplogroups (Y1, Y2, both taurine, and Y3 zebu) emerged"
I think this is a bit of a misnomer. These haplogroups are not exclusive to these cattle types. There are clearly Taurine animals in Y3, the circles are not just orange and red. Unless I am confusing the definitions (which need clearer delineation in Fig 1).

Line 161-167: "Phylogenetic analyses suggested..."
This section should probably be in the Discussion section.

Line 177: "The NJ tree shows that ancient samples show the greatest affinity for the East Asian taurine group, which consists of Tibetan, Hanwoo and Japanese cattle (Fig. 4a). The outgroup-f3 statistics and D statistics also confirmed that ancient cattle shared most derived polymorphisms with this group"

The NJ tree is harder to see, but these results do not place Tibetan cattle as close as the authors suggest, while the f3 makes clear that European cattle (north and south) are close. This requires some explanation from the authors. Perhaps Tibetan cattle have undergone too much drift which distorts the f3? Or is there a demographic explanation for the timing of the spread of the East Asian ancestry type across the Asian continent. Whatever the reasoning, the results in the figures do not match the text.

Line 184: "The earliest cattle population (East Asian taurine) was introduced from Southwest Asia before ~3,900 YBP"

Why do the authors suggest Southwest Asia specifically? Is there other evidence (archaeological evidence perhaps) the authors are using?

Line 197: "The comparisons showed that Chinese indicine and *Bos javanicus* (banteng) shared the most SNPs (~4.5 M)"

I assume the authors mean they shared derived alleles? Or do they mean that the site was segregating in both species?

Line 204: "which was a remarkably significant value"

I think "highly significant" would be more appropriate.

Line 207: "the results suggested that Chinese indicine was possibly introgressed from *Bos javanicus*"

Please be more specific than "possibly".

Figure 5 a b and c need more explanation. Which species or species' genomes are these frequencies referring to. If we are looking at an introgressed segment, why does the javanicus/yak frequency go down? I also do not know what the colours of the boxes mean in d, e and f. Please expand the legend to make clearer what is being presented.

Line 226: "defined by the smallest exogenous segment that was shared for each region showing introgressions in at least 1% of the investigated haplotypes"

I have trouble following this definition. Please make clearer.

Line 237: "One introgressed region included the well-known coat colour gene ASIP, which has been implicated as a strong candidate gene that controls coat colour"

Were there any relevant functional mutations in these regions for this gene?

Line 267: "Our results also revealed that introgression cannot explain the difference in divergence time estimates with MSMC (Supplementary Note 6 and Supplementary Fig. 17), which was consistent with the result for modern humans"

It is completely unclear what this statement is referring to above. Please clarify. It seems a to not follow from any results in the sentence above.

Line 272: "The construction also revealed a bottleneck approximately 10 kya, which could be interpreted as a signal of cattle domestication (Fig 6a)."

These MSMC plots need confidence intervals to make sense of any bottlenecks, particular for more recent periods.

Fig 6b would be easier to look at if you split within Taurus and within indices plots on a different figure to between plots.

Line 293: "In our study, Africa taurine ancestry was absent in our study"

What is mean by absent here? That there was simply none-observed that has previously been seen (it seemed that there were African taurine cattle include no?), or that by applying your method it no longer exists and you can better define African taurine ancestry as a mixture of Eurasian types?

Line 296: " In addition to two Middle East/Europe ancestries (European taurine (Y1) and Eurasian taurine (Y2a))"

I did not see any Middle Eastern samples in this study. In addition, Asian cattle from the Northwest clearly belong to this Eurasian ancestry as well. Actual central and west Asian samples would be very useful to make this ancestries clearer. Is no public data available?

Line 325: "Our work supports that ancient South China's contact with cradles of zebu domesticates from two directions: overland through Tibet and Yunnan to the southwest and by sea from Southeast Asia to its eastern coast⁴³ at least 2.9 kya, which overlapped with the appearance of zebu in South China"

I'm unsure what results these conclusions are drawn from and how they relate to *Bos javanicus* introgression. Please clarify.

Supplementary Note 1:

Line 26: "sequencing coverage was approximately 11.67X (ranging from 8.87 to 36.85)"

Given the variable coverage, it is important to check that missing heterozygosity in lower coverage samples is not affecting downstream analyses such as PCA, ADMIXTURE and NJ trees. Therefore I suggest analysis are repeated that where each sample is made pseudo-haploid to ensure no bias in the results. Alternatively, GL-aware analyses such as NGSadmix should be applied that take into account genotype uncertainty.

Line 45: "Ascertained in other species within Bovini and phased"

Similarly, I potentially worry about the effect of imputing and phasing the lower coverage samples.

Line 124: "We used BEAGLE to infer the haplotype phase and impute missing alleles"

Why is phasing being performed for the Y chromosome?

Line 214: "used software KING to estimate kinship coefficients between all the 215 individuals"

Why was this done? In addition, KING is not likely to perform well with such low coverage genomes.

Line 221: "To further investigate the gene flow between Shimao cattle and different worldwide populations, we removed "all LD" using the --indep-pairwise 50 5 0.2 option in PLINK. The f_3 statistics for (Gir; Ancient, population B) were quantified for a set of 40 worldwide populations using ~27 M SNPs."

This is a large number of SNPs given SNPs in LD was removed. Please check.

Line 248: "The nucleotide diversity (π) of all groups were calculated using a sliding window approach"

How were difference in coverages accounted for?

Supplementary Figure 7: The 5' 3' damage patterns are very messy. Why are they higher at the 3' position? They are also not very smooth, most plots are smoother than this with a nice exponential decay pattern. Please repeat this analysis using MapDamage so we can see the difference between C>T and G>A changes. I do not know what the colors being show here represent.

*****

REVIEWS

*****

**[Comment of Reviewer #1:]**

**This is a very interesting study that brings new information to understand the complex**
**history of cattle domestication. Even if it focuses on East Asian breeds, it gives a quite**
**comprehensive view of the events that occurred in this region (diversity of wild stocks**
**contributing to domestication, migrations, introgressive adaptation) and supports the**
**importance of such mechanisms in shaping modern cattle. Excepting a few restrictions**
**(see below) the study is based on a reliable sampling with regards to the question**
**addressed. The results appear to be robust as they are generally supported by several**
**analyses converging in supporting the same conclusion. This study should be of interest**
**for a wide audience.**

*Response: We thank the reviewer for the kind words, and taking the time to provide feedback.*

**[Comments of Reviewer #1:]**

**1.Main concerns.**

**My main concern is about the introgression from *Bos javanicus* in Chinese indicine**
**cattle. First, I do not challenge the occurrence of such hybridization that has already**
**been reported (e.g., Hartati et al. 2015, doi.org/10.1186/s12863-015-0229-5), but I wonder**
**how the sampling might affect the results obtained. Only two bantengs were used as**
**representative of the species, and they were from a zoo. First this is a really low sample**
**size that might not represent well the species even more if the 2 individuals are closely**
**related. Information on relatedness and inbreeding should be given. Moreover, we know**
**that hybridization between close species might occur in captivity, and this could impact**
**the result obtained. The authors should provide more information on these**
**samples/genomes and discuss how this can impact the results. The same problem is for**
***Bos gaurus*, but with less impact as no introgression from this species occurs.**

*Response: Thanks for the valuable comments. Following your suggestions, we first used the*
*KING software³⁰ to evaluate the relatedness of samples within each Bos genus species,*

including gaur, bison, wisent, banteng, and yak. The two bantengs were not close relatives
and were not within a 3rd-degree relationship (Kinship = 0.0072). Negative kinship
coefficients estimates indicated that the other *Bos* genus samples were unrelated. This part
has been added in Supplementary Note 5.

So we next explored how sample size can impact the results of introgression analysis,
especially with only two individuals. Although we cannot directly evaluate whether two
bantengs are sufficient, we can estimate the effects of sample sizes on the detection of yak
introgression to taurine cattle since we have 13 yaks available. In our study, a total of 4,238
introgressed segments (~246 Mb) were detected using the 13 yak reference. If two randomly
selected yak individuals were used for introgression analysis (totally 78 combinations from 13
samples), the detection ratio of introgressed segments is >99% for highly frequent segments
(ie., \geq two alleles in nine Tibetan cattle) and can still reaches 75% for low frequent ones (one
allele in nine Tibetan cattle). In our study, the mean proportion of the genome inferred to be of
*Bos javanicus* ancestry was 2.92 ± 0.45 %, with 2.61 ± 0.42 % showing at least two alleles.
So we believe that our two banteng samples should be sufficient for the detection of
introgressed segments, especially for the highly confident regions with at least two shared
alleles in indicine cattle. In addition, only introgressed segments that shared at least two
alleles in Chinese indicine groups were used for the functional enrichment and introgression
time analysis.

We have added our simulation results in the Supplementary Note 5.

**[Comments of Reviewer #1:]**

**A second point is about the adaptive aspect of the introgression. In the title and abstract,**
**the authors refer to 'adaptive' introgressions. However this is speculative, as the genes**
**found in introgressed regions are not even related to environmental changes. To confirm**
**such adaptive process for at least some genes, the authors should link the presence of**
**banteng-specific alleles in cattle to the occurrence of given environmental conditions.**
**Otherwise, they cannot claim that this is an adaptive process.**

**Response:** *Thanks for the valuable comment and we apologize that we did not describe the*
*'adaptive' introgressions clearly.*

*In fact, we have some examples. The GO analysis identified a significant*
*over-representation of genes of introgressed segments involved in disease resistance and*
*response to hypoxia, which probably have contributed to the adaption of extreme*
*environmental conditions of the Tibetan Plateau. We highlighted the EGLN1 gene*
*introgressed from yak, a gene for the hypoxia-inducible factor (HIF) pathway that has been*
*repeatedly identified as including targets for selection to high-altitude adaptations in Tibetan*
*humans*³².

*Bos javanicus were well adapted to the tropical climate environment, food, and local*
*pathogens. The introgressed genes from banteng to Chinese indicine involved in sensory*
*perception (bta04740, olfactory transduction; bta04742, taste transduction), which will help*
*Chinese indicine adapt to the food of tropical climate environment.*

*Following this suggestion, we also retrieved several genes for thermotolerance in*
*banteng introgressed segments and hypoxia genes found in yak introgressed regions to*
*support adaptive introgression. We have added the information in the manuscript in Result*
*Section and Supplementary Note 5, Supplementary Table 30,31 and Supplementary Fig.*
*19,20.*

*Text has been added in the manuscript as follows:*

*“Of these, ten nonsynonymous substitutions were present within the T2R12, TAS2R9,*
*and TAS2R6 genes. These genes are functionally relevant to bitter taste in humans and giant*
*pandas*^{1,2} *and may have a similar role in Chinese indicine cattle. We also identified several*
*introgressed genes that favor local adaptation to tropical climate environments in Chinese*
*indicine cattle (Supplementary Table 30 and Supplementary Fig. 19). For example, we found*
*several heat shock protein (HSP) family-related genes, including, HSPA1A, HSPB8, HSPA8,*
*HSPA4, HSPB2, and HSF2, which are involved in the key cellular defense mechanisms during*
*exposure in hot environments*²⁸. *We found several genes related to hair cell differentiation*
*and blood circulation, including ATOH, GNAI4, VPS13, and KIF2B, which also play an*
*important role in the temperature adaptation of Chinese pigs*²⁹.

*In addition, we were able to retrieve several hypoxia genes in introgressed regions*
*that may help Tibetan cattle adapt to hypoxic environments (Supplementary Table 31 and*
*Supplementary Fig. 20). These included candidate genes COPS5, IL1A, IL1B, MMP3, and*

*EGLN1*, for the hypoxia-inducible factor (HIF) pathway that has been repeatedly identified
among targets for selection to high-altitude adaptations in Andeans, Tibetans, and yaks³²⁻³⁴.
We also observed two genes, *RYR2* and *SDHD*, involved in mediating calcium homeostasis
that regulate the response to hypoxia³⁵”

**[Comments of Reviewer #1:]**

**2. Other concerns.**

**A global concern is about Figure and Table numbering. In both the main text and**
**supplementary material the Fig and Tab numbers should be checked and modified when**
**necessary. It is sometimes difficult to find the good information as the item number do**
**not correspond : e.g. suppl. Table 5 refers to the PCA not to SNPs (line 135), Suppl.**
**Table 8 do not exist, etc. ; Suppl. Table 7 (line 172) should correspond to Suppl Table**
**10; also Figure 3 is called before Figure 2 in the main text. Figures would be more easily**
**interpreted by the reader with a few more information, such as for example the**
**indicine/taurine origin of the breed in Fig. 1, Fig. 6a, Suppl. Figures 1-5.**

**Response:** *We apologize that we did not describe the number of Figures and Tables clearly.*

*We have double-checked the Figure and Table numbering.*

*All information has been added in the Figures as below:*

*In Fig. 1, Supplementary Figs 1, 2, 3, 5, and 6, we have added different shapes to*
*represent different origins of cattle. “□” represent *Bos taurus*, “○” represents *Bos indicus*, “×”*
*represent *Bos taurus* × *Bos indicus* hybrid.*

*In Fig. 6a, we have added information for five breeds. BTA for taurine origin*
*cattle(Hereford, Gelbvieh, and Tibetan) , BIN for indicine origin(Wannan, Haryana and*
*Sahiwal).*

*In Supplementary Fig. 4, we have added “Breeds 1-13,15-22 are taurine origin; 37-44*
*are indicine origin; Others are hybrid origin.” to explain the origin of different breeds.*

*In Supplementary Fig. 7, we have added “Y1 and Y2” belong to *Bos taurus*, Y3 belongs to*
**Bos indicus*.*

*In Supplementary Fig. 8, we have added “T1,T2,T3,T4 belong to *Bos taurus*, I1, I2*
*belong to *Bos indicus*.”*

**[Comments of Reviewer #1:]**

**More details should be given for some methods :**

**suppl. note 1 - It is not clear weather the genotype data imported from public databases**
**were used for variant discovery or not. The set of genomes used for variant discovery**
**should be stated (as it has an impact on the polymorphism revealed). What kind of**
**information was retrieved from the databases ?**

***Response:** We apologize for the confusing description. Yes, we used both 146 public available*
*cattle samples, and 114 Chinese and Indian samples for variant discovery.*

*We totally used three sample sets in this manuscript as bellow.*

*For the first set, we sequenced 114 cattle samples. The average sequencing coverage*
*was approximately 11.88 X (ranging from 8.29 X to 36.85 X) per individual. For the second*
*set, we retrieved 146 public available from 23 cattle breeds worldwide. The sequencing*
*coverage was approximately ~13.34 X (ranging from 3.92 X to 25.26 X) per individual. The*
*third set included 26 whole genome data from extant wild species of Bovini, including gaur,*
*bison, and wisent, banteng, gayal, and buffalo. They were used for outgroup and introgression*
*analysis^{1,7-10}. The sequencing coverage of those wild species was approximately ~11.17 X*
*(ranging from 4.12 X to 38.96 X) per individual.*

*We have added this information to the Supplementary Note 1.*

**[Comments of Reviewer #1:]**

**Suppl. note 2 - it is not clear if all fourfold-degenerate sites were chosen or if it is a**
**subsample; the distance used for building the NJ tree should be given; information on**
**how was build the ML tree should be provided.**

***Response:** We apologize that we omitted the detail of this method. In the previous manuscript,*
*we extracted 5,194,125 fourfold-degenerate sites, after filtering the missing sites, a total of*
*4,021,677 sites were used for building the NJ tree. Following the Reviewer's suggestions, we*
*have constructed a new NJ tree using all of the autosomal sites (60,449,904) to replace the*
*previous NJ tree built by the fourfold-degenerate sites (see revised Fig. 1). The topology of*
*these two trees is generally the same. We also added the distance information used for*

building the NJ tree in Supplementary Table 7. The ML tree was built using the ML approach
implemented in TreeMix (Supplementary Fig. 4). Detailed information regarding how to build
the ML tree has been added to the Supplementary Note 2 as below:

*We also inferred a population-level phylogeny using the ML approach implemented in*
*TreeMix. The window size of 1000 was used to account for linkage disequilibrium (-k) and*
*“-global” to generate the ML tree.*

**[Comments of Reviewer #1:]**

**The interpretation linking the bottleneck inferred from MSMC analyses at 10 kya to**
**domestication (273) is not straightforward as the low N_e at this time follow a continuous**
**demographic decline that began at least 20 kya before.**

*Response: Thanks for the suggestion. It has been revised as “For both zebu and taurine cattle,*
*a dramatic decline in effective population size (N_e) was detected 20~30 kya, which likely*
*reflects the major climatic change at the end of the last glacial maximum (LGM)³ (Fig. 6a),*
*predating cattle domestication. Results also confirmed a N_e decay following the onset of*
*domestication.”*

**[Comments of Reviewer #1:]**

**3. Minor points:**

**Main document**

**Line 50. Here 'aurochs' are actually Indian aurochs (ie indicine subspecies). Should be**
**clearer if explicit I think.**

*Reponses: Thank you. It has been revised as “Indian aurochs”.*

**[Comments of Reviewer #1:]**

**Line 99. I would find 'analysis of the genetic structure' more convenient that**
**'population genetic...'**

*Reponses: Thank you. It has been revised as “population genetic”.*

**Line 204. remove 'remarkably'**

**Reponses:** *Changed as suggested.*

**[Comments of Reviewer #1:]**

**Lines 293-295. The sentence is not clear, please rephrase.**

**Reponses:** *Thanks for your comments. Following your suggestions, it has been revised as*

*“Previous population analyses based on nuclear markers have confirmed genetic ancestry of*

*European, African, Indian, and American cattle⁴⁻⁷. However, the histories of East Asian cattle*

*populations based on the genome level have been poorly understood.”*

**[Comments of Reviewer #1:]**

**Line 306. I think that 'areas' would be better adapted than 'sites'**

**Reponses:** *It was corrected as suggested.*

**[Comments of Reviewer #1:]**

**Lines 324-328. The sentence is not clear, please rephrase.**

**Reponses:** *We apologize for the confusing description. It has been revised as*

*“The *Bos javanicus* species historically ranged throughout the southeast mainland and*

*southern China⁴⁸. We observed that Chinese indicine inherited ~2.92 % genome component*

*from *Bos javanicus* ancestry at least 2.9 kya, which was consistent with the earliest evidence*

*for zebu in China 3 kya^{14,15}. The initial period of introgression is most likely due to migrating*

*eastward zebu reached China⁴⁹.”*

**[Comments of Reviewer #1:]**

**Lines 346-347. Would be clearer if reminding here that the introgression in Tibetan**

**taurine is from Yak.**

**Reponses:** *Thanks for comments. It has been revised as “Similar introgression was also found*

*from yak into Tibetan taurine.”*

**[Comments of Reviewer #1:]**

**Line 351. 'improve the variability'. Do you mean 'increase'?**

**Reponses:** *Thanks for comments. It has been revised as “increase the variability”*

**[Comment of Reviewer #1:]**

**Supplementary Material**

**Lines 45-49. Not clear. Please rephrase**

**Reponses:** *We apologize that we did not describe the method clearly. It has been revised as*
*“The whole genome data from 7 extant wild species were mapped in the same way. We used*
*60.4 million SNPs as reference list to genotype the combine set of 260 samples and 7 extant*
*wild species. The final genotype data were imputed and phased using BEAGLE (version*
*4.1).”*

**[Comments of Reviewer #1:]**

**Line 200. what do you mean by 'genetically pure' individuals ?**

**Reponses:** *Apologize for the ambiguous sentence. It has been revised as “we selected "core"*
*groups of the five components based on the structure according to ADMIXTURE for the*
*phylogenetic analysis.”*

**[Comments of Reviewer #1:]**

**Lines 270-272 and 272-275. These two sentences are not really clear. Please rephrase**

**Reponses:** *We apologize for the confusing description. It has been revised as “Then, 5-kb*
*sliding windows were used to calculate the mean frequency of banteng and zebu alleles in the*
*Chinese indicine group and the frequency of two type of alleles were plotted (Supplementary*
*Fig. 17).”*

**[Comments of Reviewer #1:]**

**- line 336. 'individual' -> individual**

**Reponses:** *It was corrected as suggested.*

**[Comments of Reviewer #1:]**

**Suppl. Table 22. I wonder why many contiguous segments were considered**

**independently and not as a single region (e.g., chr 1 6336595 -10025102,**
**10025102-10327999, 10327999 -110722931 , etc.). Is it related to previous haplotype**
**definition?**

***Reponses:*** *We did not merge the contiguous segments, because the contiguous segments is*
*actually from different individuals, leaded by historical recombination.*

**[Comment of Reviewer #2:]**

**Chen et al. analyzed whole-genome resequencing data of hundreds of cattle, including 8**
**ancient cattle from East Asia. They classify cattle into 5 groups. They see a split in Asian**
**taurine cattle and a split in indicine cattle. They estimate that the split between Chinese**
**indicine and Indian indicine occurred approximately 36 to 49 kya. They also note**
**banteng and yak introgression into Asian cattle.**

**The authors claim that the histories of East Asian cattle are poorly understood. However,**
**some of the events they describe have been previously reported. Banteng introgression**
**into Chinese indicine has previously been reported by Decker et al. 2014 (Chinese**
**Hainan and Luxi breeds).**

***Response:** Thank you for valuable comments and suggestions. Following your suggestions,*
*we described and cited that Banteng introgression into Chinese indicine have been reported*
*by Decker et al. and we have rephrased our abstract and introduction part to make our*
*descriptions more precise. It has been revised as “the complex histories of East Asian breeds*
*need more deciphering.” in the Abstract. And previously reports were added in the*
*Introduction part. It has been added in the second part of Introduction:*
*“Based on genomic SNP array data, previous studies provided evidence of introgression*
*within the Bos genus, such as banteng introgression into the Chinese Hainan breed and*
*bovine introgression into Mongolian yaks in East Asia^{6,16}. All these contribute to the complex*
*histories of East Asian cattle.”*

**[Comments of Reviewer #2:]**

**I also suspect of the divergence date between Indian and Chinese indicine cattle. The fit**
**between the dadi model and the observed data doesn't look great. The model has the**
**weakest fit for variants that are at high frequency in the Chinese samples and moderate**
**frequency in the Indicine samples. It also has a weak fit for rare variants in Indian**
**samples that are at a moderate frequency in Chinese. The Wannan have the highest**
**effective population size in the MCMC results. However, in the dadi model, they are**
**estimated to be smaller than the Indian indicine Ne.**

***Response:** Thank you for pointing out this problem. We have improved and simulated an*

*isolated-migration model with the same dataset under the two-population model in $\partial a \partial i$*
 *independently. Regarding the effective population size, the $\partial a \partial i$ model were used to simulate*
 *recent demographic fluctuations. The estimated effective population size from 40 kya to 10*
 *kya in the MSMC and $\partial a \partial i$ were similar. See Fig. 6 and Supplementary Note 6 for the updated*
 *results.*

 *Fig. 6 (c) $\partial a \partial i$ result showing the divergence time of Chinese indicine and Indian indicine.*
 *The ancestral population is in grey, Chinese indicine in red and Indian indicine in orange.*
 *The width shows the relative effective population size. The model also supported low level*
 *gene flow indicated by arrows.*

 *Supplementary Figure 22. Comparisons of allele frequency spectra (AFS) between the model*
 *and real data of Chinese indicine and Indian indicine populations using $\partial a \partial i$. (a) Marginal*
 *AFS of the real data for each pair of populations. (b) AFS of the maximum-likelihood model*

*simulated based on the real data. The residuals between the model and real data are shown in*
*heat maps (c) and bar graphs (d).*

**[Comments of Reviewer #2:]**

**On page 15, lines 299 through 302, there is a discussion of male-mediated gene flow. If**
**the females were stationary and the males were moving, wouldn't the mitochondria**
**DNA reflect geography? However, what could be occurring is that females are being**
**moved (likely from domestication centers) and introgression from yak, banteng, and**
**possibly regional aurochs are creating the differences in chromosome Y lineages?**
**Perhaps this is what the authors intended to say and I didn't follow. Regardless, this**
**section should be clarified.**

***Response:** Apologies for the ambiguous sentence. It has been revised as "Our results show*
*that paternal lineages have a clear phylogeographical structure. The nuclear genome*
*structure is mainly consistent with that of paternal lineages, but is dissimilar to that of*
*maternal lineages."*

**[Comments of Reviewer #2:]**

**On lines 317 to 320 of the Discussion, the population structure of Eurasian taurines is**
**discussed. Could trade along the Silk Road influenced the relationships between**
**Southern European (Italian, etc.) cattle and Mongolian and North-Central China cattle?**

***Response:** We agree with the reviewer that the connection between Europe and East Asia*
*could influence the relationship between Southern European (Italian, etc.) cattle and*
*Mongolian and North-Central China cattle. However we think it is more likely due to the*
*expansion of early pastoralism, because domesticated cattle are not suited to long distance*
*transportation in arid desert. The corresponding discussions have been revised: "The*
*Eurasian strand possibly was introduced and spread later by the expansion of early*
*pastoralism, leading to west-to-east immigrations, which may explain the similar component*
*of Southern European cattle and Mongolian cattle. In addition, the recent introgression of*
*Mongolians into huge and extensive areas of China may also lead to a gene flow across*
*eastern Eurasia and partial replacement of East Asian taurine (or Y2b lineage) in northern*

*China.*”

**[Comments of Reviewer #2:]**

**Additional comments:**

**Line 266. Why is mutation rate reported per year?**

*Response: Apologies for the ambiguous description. It has been revised as “The multiple*
*sequentially Markovian coalescent (MSMC) approach*³⁶ *with a generation time of $g = 6$ and*
*a mutation rate per generation $\mu_g = 1.26 \times 10^{-8[24,25]}$ was used to reconstruct the population*
*history of the five core groups.*

**[Comments of Reviewer #2:]**

**Line 285, misspelling of Indian.**

*Response: It has been corrected as suggested.*

**[Comment of Reviewer #3:]**

**General Comments:**

**In general this is a good paper that in my opinion is well worth publishing in a journal**
**such as Nature Communications subject to the revisions/clarifications described below. I**
**enjoyed reading it. Though new methods are not developed, existing ones are applied**
**appropriately, and a lot of new data is generated. The conclusions are sound and**
**supported by the data, and interesting from the perspective of researchers trying to**
**understand the process of domestication (the parts on introgression are particularly**
**interesting).**

**My comments below are meant primarily to help improve the reading of the manuscript,**
**especially for non-cattle experts such as myself, as sometimes this is not clear.**

*Response: Thank you for your time spent on reviewing our manuscript. We sincerely*
*appreciate your high evaluation of our findings. Those comments are all valuable and very*
*helpful for revising and improving our paper.*

**[Comment of Reviewer #3:]**

**The only two disappointing things from the study are a) the massive gap in sampling**
**between Europe and China (there is nothing from Central Asia), which would help**
**better contextualize the results, especially for the “Eurasian” component, and the very**
**limited analysis of the ancient DNA, which to me is not fully exploited (how did some**
**functional alleles look for example?). If there is Central Asian data is available, I would**
**strongly urge the authors to include this.**

*Response: Thanks for the reviewer’s comments and suggestion. In our study, Kazakh cattle*
*were sampled from the border of the Northwest part of China and Kazakhstan, which can be*
*taken to represent Central Asian samples. In addition, we have further added the whole*
*genome data of eight female Rashiki cattle from NCBI, which are sampled from Iran in the*
*Middle East. The Rashiki cattle represented hybridization of taurine and indicine. And*
*population genetic structure is consistent with our previous results. Thus these data could*
*partially fill the gap in sampling between Europe and China. Unfortunately, the available*
*sample of other Central Asia areas is very limited. It is not feasible for us to It is not feasible*

*for us to collect pure (meaning not mixed by outside blood) male samples from these areas in*
*within the time frame of this study. In the future, we will try our best to collect such samples*
*for further analysis. We have added the result in our manuscript.*

*Regarding the functional alleles of our ancient samples, our main focus of this*
*manuscript is to explore the genome origin of cattle from East Asia. The exploration of the*
*domestication genes of ancient samples through searching for selective sweeps would be*
*another highly involved study. We think that our work about the modern and ancient samples*
*will be a useful resource for other researchers for such analysis and we agree with that it*
*could be interesting for further studies.*

**[Comment of Reviewer #3:]**

**Specific Comments:**

**Line 64: Sentence beginning: “Nuclear markers confirmed genetic ancestry”**

**Could do with a little more detail on what the patterns were in these previous studies (i.e.**
**existing knowledge of cattle genetic structure. Similarly, a little explanation on the**
**background/known differences between taurine (taurus) and zebu (indicus) are needed**
**for non-specialists**

**Response:** *Thanks for the comments. Following your comments, we have added the*
*background differences of taurine and zebu cattle and details of the world patterns. In the*
*Introduction part, this part has been revised as below:*

*“Two primary areas of domestication in the Near East and the Indus Valley resulted in*
*humpless taurine (*Bos taurus*) and humped zebu (*Bos indicus*) cattle, respectively³. Generally,*
*indicine cattle can withstand high temperatures compared with taurine breeds. Population*
*analyses based on genomic SNP array data revealed three major groups: Asian indicine,*
*Eurasian taurine, and African taurine; and also recovered the historical migratory routes of*
*cattle from their centers of origin across the world⁴⁻⁷. However, the whole genome diversity of*
*cattle from East Asia has not been investigated in depth.”*

**[Comment of Reviewer #3:]**

**Line 74: “However, several ancient Chinese aurochs samples have yielded a highly**

**divergent haplogroup, implying a minimal local matrilineal incorporation and complex**
**histories of East Asian cattle.”**

**Not sure what the link is here with the previous statement, and how the ancient data**
**reflect complex histories.**

*Response: Thank you for your suggestion. We agree with your suggestion. We have deleted*
*this sentence.*

**[Comment of Reviewer #3:]**

**Fig 1 needs more information for interpreting the results more easily. Which points are**
***Bos taurus* and which are *Bos indicus* (I assume red and orange but had to guess). Also,**
**why is yellow considered southwest China and red south China,**

*Response: Thank you for your suggestion. We have added more explanation to the legend of*
*Fig. 1. Different shapes were used to represent *Bos taurus* and *Bos indicus* and hybrid with*
*different colours representing their geographical distribution. We also assigned the southwest*
*samples to South China group.*

*Figure 1 Population structure and relationships of East Asian cattle compared with those of*
*other cattle.*

*(a) Geographical origins of cattle breeds. Breed name associated with each number is listed*

*in Fig. 2a. Principal component analysis (PCA) showing PC1 against PC2 (b) and PC1*
*against PC3 (c). (d) A neighbour-joining (NJ) phylogenetic tree constructed using*
*whole-genome SNPs data. The scale bar represents pair-wised distance of different*
*individuals. Colors reflect sampling locations.*

**[Comment of Reviewer #3:]**

**Line 95: “Among these genomes, we detected a total of 60.4 million putative autosomal**
**SNPs”**

**What is the importance of this, especially given the uneven coverage?**

*Response: We agree with the reviewer that we should not emphasize its importance. This part*
*has been revised as “A total of 60.4 million putative autosomal SNPs were identified and used*
*in subsequent analysis.”*

**[Comment of Reviewer #3:]**

**Line 102: “and the second component was driven by a split of East Asian cattle from**
**other individuals”**

**Is this really the case? The major split appears to be between India/Pakistan/African**
**versus everyone else. East Asian cattle show some structure but this isn’t surprising**
**given there are many more of them than elsewhere (sample number has an influence on**
**PCA structuring due to increased covariance amongst samples).**

*Response: We apologies that we might have misled the reviewer. This part has been revised as*
*“For Bos taurus, a separation was also found between European and East Asian taurine*
*cattle along the second component (Fig. 1b and Supplementary Fig. 1). Within Bos indicus, a*
*clear partitioning was apparent between cattle from India and South China (Fig. 1c and*
*Supplementary Fig. 2).”*

**[Comment of Reviewer #3:]**

**Line 105: “Southwest China cattle exhibited intermediate ancestry”**

**They seem closer to south china than India/Pakistan**

*Response: We agree with the reviewer. This sentence has been moved to the part of result of*
*ADMIXTURE and revised as “Dianzhong cattle in South China is composed of hybrid*

*Indian-Chinese indicine genotypes.” in line 123.*

**[Comment of Reviewer #3:]**

**Line 106: “A separation was also found between European and East Asian taurine**
**cattle along the third component”.** I do not see this, European cattle seem to have the
**same PC3 co-ordinates as NC and SW China.**

***Response:** Thank you for your comments. The previous Fig. 1b could not clearly demonstrate*
*the separation between European and East Asian taurine cattle. We have added the different*
*shapes to represent the Bos taurus, Bos indicus, and hybrids. Following the comments from*
*you and other reviewers, we have repeated the PCA result using Eigensoft software and the*
*genotype likelihood approach PCA-ANGSD. The results are updated in Fig. 1b,1c and in the*
*Supplementary Fig. 3, which now can show the separation clearly. This sentence has been*
*revised as “For Bos taurus, a separation was also found between European and East Asian*
*taurine cattle along the second component”.*

**[Comment of Reviewer #3:]**

**Line 108: “The same population affinities were recovered in trees constructed by the**
**neighbour-joining (NJ) method”**

**This structure is very hard to see as the branch lengths are so small. Consider removing**
**to supp and making bigger.**

***Response:** Thanks a lot for your suggestion. We used another NJ tree built using all*
*autosomal sites (60,449,904) to update the NJ tree instead of using fourfold-degenerate sites.*
*The NJ tree of all autosomal SNPs showed a more clear structure with longer branch lengths.*
*Please see Fig.1.*

**[Comment of Reviewer #3:]**

**Lines 107-129: Why is Fig 3 not presented as Fig 2 in terms of text order? In general I**
**think this section needs a little bit of cleaning up and perhaps shortening. There is some**
**repetition with regards to the grouping.**

***Response:** This part has been revised and we have removed the repetition about the grouping.*

*We have adjusted the order of Fig. 2 and Fig. 3. In addition, we have double-checked all*
*Figure and Table numbering.*

**[Comment of Reviewer #3:]**

**Line 136: “three common Y haplogroups (Y1, Y2, both taurine, and Y3 zebu) emerged”**

**I think this is a bit of a misnomer. These haplogroups are not exclusive to these cattle**
**types. There are clearly taurine animals in Y3, the circles are not just orange and red.**

**Unless I am confusing the definitions (which need clearer delineation in Fig 1).**

***Response:** We apologies that we might have misled the reviewer. Previous studies of*
*Y-chromosomal variation identified two *Bos taurus* (taurine) haplogroups (Y1 and Y2; both*
*composed of several haplotypes) and one *Bos indicus* (indicine/zebu) haplogroup (Y3) (PLoS*
*One. 6(1): e15922 (2012); Heredity 105: 511–519 (2010)). In addition, we added more*
*explanation to the legend of Fig. 1. We used different shapes to distinguish the *Bos taurus*,*
**Bos indicus*, and their hybrids. So three common Y haplogroups (Y1, Y2, both taurine, and Y3*
*zebu) identified in previous studies were consistent with the origins of different cattle. The*
*cattle from North-Central China have hybrid origin. So they may have both taurine and*
*indicine haplogroups.*

**[Comment of Reviewer #3:]**

**Line 161-167: “Phylogenetic analyses suggested...” This section should probably be in**
**the Discussion section.**

***Response:** Following your suggestion, we have moved and integrated this into the first*
*paragraph of the Discussion:*

*“Previous population analyses based on nuclear markers have confirmed genetic*
*ancestry of European, African, Indian, and American cattle⁴⁻⁷. However, the histories of East*
*Asian cattle populations based on the genome level have been poorly understood. The*
*following five major ancestries (and paternal lineages) were consistently observed in our*
*study. In addition to European taurine (Y1) and Indian indicine (Y3b), three types of*
*ancestries, including Eurasian taurine (Y2a) and two distinct ancestries (East Asian taurine*
*(Y2a) and Chinese indicine (Y3)), were observed in East Asian cattle. Phylogenetic analyses*

*suggested that present-day East Asian domesticated cattle mainly originated from three bull*
*populations. The north-western populations shared the Y2a sub-haplogroup with the*
*Central-South Europe populations, thus representing Eurasian taurine ancestry, whereas the*
*Y2b sub-haplogroup was predominant in cattle from Northeast Asia and Tibet, thus*
*representing East Asian taurine ancestry. Cattle in South China are primarily of *Bos indicus**
*ancestry and belong to the Y3a sub-haplogroup (Fig. 2b and 2c), and they also carried a*
*maternal IIa lineage. Our results show that paternal lineages have a clear phylogeographical*
*structure. The nuclear genome structure is mainly consistent with that of paternal lineages,*
*but is dissimilar to that of maternal lineages.”*

**[Comment of Reviewer #3:]**

**Line 177: “The NJ tree shows that ancient samples show the greatest affinity for the**
**East Asian taurine group, which consists of Tibetan, Hanwoo and Japanese cattle (Fig.**
**4a). The outgroup-f3 statistics and D statistics also confirmed that ancient cattle shared**
**most derived polymorphisms with this group”**

**The NJ tree is harder to see, but these results do not place Tibetan cattle as close as the**
**authors suggest, while the f3 makes clear that European cattle (north and south) are**
**close. This requires some explanation from the authors. Perhaps Tibetan cattle have**
**undergone to much drift which distorts the f3? Or is their a demographic explanation**
**for the timing of the spread of the East Asian ancestry type across the Asian continent.**
**Whatever the reasoning, the results in the figures do not match the text.**

***Response:*** *Yes, you are correct. Tibetan cattle is indeed not as close to ancient samples as*
*other East Asian samples. This part has been revised as “The NJ tree reveals that ancient*
*samples show the closet affinity for the Hanwoo and Japanese cattle (Fig. 4a). The*
*outgroup-f3 statistics and D statistics also confirmed that ancient cattle shared most derived*
*polymorphisms with Northeast Asian cattle and Japanese cattle, respectively (Fig. 4b,*
*Supplementary Fig. 11, and Supplementary Table 18). The Tibetan cattle were divergent from*
*the other East Asian cattle earlier or undergone a stronger drift after the separation might*
*result the positive D scores in Tibetan cattle.”*

**[Comment of Reviewer #3:]**

**Line 184: “The earliest cattle population (East Asian taurine) was introduced from**
**Southwest Asia before ~3,900 YBP”**

**Why do the authors suggest Southwest Asia specifically? Is there other evidence**
**(archaeological evidence perhaps) the authors are using?**

*Response: Sorry for the ambiguous sentence. We have added a little background information*
*of the cattle domestication in the Introduction part: “Two primary areas of domestication in*
*the Near East and the Indus Valley resulted in humpless taurine (*Bos taurus*) and humped*
*zebu (*Bos indicus*) cattle, respectively³”. So we speculate that the earliest cattle population*
*(East Asian taurine) might be introduced from the Southwest Asia domesticated area.*
*Considering there was no evidence that our earliest cattle were directly introduced for*
*Southwest Asia, this part has been revised as “We speculate that the earliest cattle population*
*(East Asian taurine) might be introduced before ~3,900 YBP, whereas an exotic introgression*
*(Eurasian taurine), which is now prevalent in the middle part of China, resulted from a*
*second migration event.”*

**[Comment of Reviewer #3:]**

**Line 197: “The comparisons showed that Chinese indicine and *Bos javanicus*(banteng)**
**shared the most SNPs (~4.5 M)”**

**I assume the authors mean they shared derived alleles? Or do they mean that the site**
**was segregating in both species?**

*Response: Sorry for the ambiguous sentence. It has been revised as “The comparisons*
*showed that Chinese indicine and *Bos javanicus* (banteng) shared the most derived alleles*
*(~4.5 M), followed by *Bos frontalis* (gayal) and *Bos taurus* (Supplementary Table 19).”*

**[Comment of Reviewer #3:]**

**Line 204: “which was a remarkably significant value”**

**I think “highly significant” would be more appropriate.**

*Response: Thanks for pointing out this problem. It has been revised as “which was a highly*
*significant value”.*

**[Comment of Reviewer #3:]**

**Line 207: “the results suggested that Chinese indicine was possibly introgressed from**
***Bos javanicus*”**

**Please be more specific than “possibly”.**

*Response: Thank you. We have added the Z-score to explain the “possibly”. It has been*
*revised as “D statistics tests were applied following the tree topology (Buffalo, Chinese*
*indicine; Bos javanicus, Bovini), and the results suggested that Chinese indicine was most*
*possibly introgressed from Bos javanicus, which produced a highly significant Z-score of*
*-51.54 (Supplementary Table 21).”*

**[Comment of Reviewer #3:]**

**Figure 5 a b and c need more explanation. Which species or species’ genomes are these**
**frequencies referring to. If we are looking at an introgressed segment, why does the**
**javanicus/yak frequency go down? I also do not know what the colours of the boxes**
**mean in d, e and f. Please expand the legend to make clearer what is being presented.**

*Response: We apologize for the confusing description. In the Fig. 5a and b, blue and red dots*
*show the relative frequencies of zebu- and banteng-specific genotypes, respectively. We also*
*have added the colour legend in the Fig. 5. And we also added the legend for different boxes.*

**[Comment of Reviewer #3:]**

**Line 226: “defined by the smallest exogenous segment that was shared for each region**
**showing introgressions in at least 1% of the investigated haplotypes”**

**I have trouble following this definition. Please make clearer.**

*Response: We apologize for the confusing description. It has been revised as “We next*
*exploited the gene content of 1,852 introgressed intervals that was shared at least two*
*investigated haplotypes (Supplementary Table 23).”*

**[Comment of Reviewer #3:]**

**Line 237: “One introgressed region included the well-known coat colour gene ASIP,**

**which has been implicated as a strong candidate gene that controls coat colour”**

**Were there any relevant functional mutations in these regions for this gene?**

*Response: Thanks for your comments. No nonsynonymous SNPs were found within the region,*
*suggesting that the potential target for selection might be regulatory mutations. We have*
*added this part in the manuscript.*

**[Comment of Reviewer #3:]**

**Line 267: “Our results also revealed that introgression cannot explain the difference in**
**divergence time estimates with MSMC (Supplementary Note 6 and Supplementary Fig.**
**17), which was consistent with the result for modern humans”)**

**It is completely unclear what this statement is referring to above. Please clarify. It seems**
**a to not follow from any results in the sentence above.**

*Response: Apologies for the ambiguous sentence. In this part, we wonder that banteng*
*introgression might result in changes in the estimated Ne and divergence time. So we*
*prepared the control data that masked the introgression regions of banteng and yak in*
*Chinese indicine and Tibetan taurine, respectively. The analysis of the control data showed*
*that the patterns of divergence time among different group and population size history did not*
*change. So we concluded that limited introgressions have no impact on demographic history*
*using MSMC. It has been revised as “We first evaluated the impaction of introgression on the*
*estimated Ne and divergence time and our results revealed that limited introgression have no*
*impact on demographic history simulation using MSMC.”*

**[Comment of Reviewer #3:]**

**Line 272: “The construction also revealed a bottleneck approximately 10 kya, which**
**could be interpreted as a signal of cattle domestication (Fig 6a).”**

**These MSMC plots need confidence intervals to make sense of any bottlenecks,**
**particular for more recent periods.**

*Response: Apologies for the ambiguous sentence. For the divergence time, the confidence*
*intervals of relative cross-coalescence analysis were the 0.25 to 0.75 range. This sentence has*
*been revised as “For both zebu and taurine cattle, a common dramatic decline in effective*

population size (N_e) was detected 20~30 kya, which likely reflects the major climatic change
 at the end of the last glacial maximum (LGM)³⁸ (Fig. 6a), predating cattle domestication.
 Results also confirmed a decline of N_e following the onset of domestication.”

**[Comment of Reviewer #3:]**

**Fig 6b would be easier to look at if you split within taurus and within indicine plots on a**
 **different figure to between plots.**

**Response:** Thanks for your suggestions, Fig. 6b. has three parts of the divergence time:

Part1: The divergence time among three taurine groups;

Part2: The divergence time between two indicine groups;

Part3: The divergence time among taurine and indicine groups;

We have tried sub-figures but we found the three divergence time would be more contrasting
 within one figure. In order to make the figure easier to follow, we have added the “BTA/BTA
 divergence” for split within taurine groups, “BIN/BIN divergence” for split within indicine
 groups, and “BTA/BIN divergence” for split between indicine and taurine groups.

**Fig 6 (b)** Inferred relative cross coalescence rates between pairs of populations over time
 based on 4 haplotypes each from Hereford, Gelbvieh, Tibetan, Wannan and Indian breeds
 (Haryana and Sahiwal). The x-axis shows time, and the y-axis shows a measure of similarity
 for each pair of compared populations.

**[Comment of Reviewer #3:]**

**Line 293: “In our study, Africa taurine ancestry was absent in our study”**

**What is mean by absent here? That there was simply none-observed that has previously**

**been seen (it seemed that there were African taurine cattle include no?), or that by**
**applying your method it no longer exists and you can better define African taurine**
**ancestry as a mixture of Eurasian types?**

***Response:** Sorry for the ambiguity. Previous population analyses based on bovine 50K chip*
*confirmed three major groups of Asian indicine, Eurasian taurine, and African taurine*
*worldwide. We have added this information into the Introduction part. But in our study, we*
*did not detect the African taurine group, perhaps due to the limited samples from Africa. We*
*have removed this sentence and the first paragraph of Discussion has been revised.*

**[Comment of Reviewer #3:]**

**Line 296: " In addition to two Middle East/Europe ancestries (European taurine (Y1)**
**and Eurasian taurine (Y2a))"**

**I did not see any Middle Eastern samples in this study. In addition, Asian cattle from the**
**Northwest clearly belong to this Eurasian ancestry as well. Actual central and west**
**Asian samples would be very useful to make this ancestries clearer. Is no public data**
**available?**

***Response:** Thank you for the comment. We apologize that we did not describe clearly.*
*In the current version, we have added eight Rashoki cattle whole genome data in the*
*manuscript, which were downloaded from NCBI. Rashoki cattle are Iran native cattle, which*
*can represent the Middle East. The Rashiki cattle showed hybridization of three types of*
*ancestries, including Eurasian taurine, East Asian taurine and Indian indicine. No further*
*public data are available.*

*This part has been revised as "In addition to European taurine (Y1) and Indian*
*indicine (Y3b), three types of ancestries, including Eurasian taurine (Y2a) and two distinct*
*ancestries (East Asian taurine (Y2a) and Chinese indicine (Y3)), were observed in East Asian*
*cattle (Fig. 2b and Fig. 2c)."*

**[Comment of Reviewer #3:]**

**Line 325: "Our work supports that ancient South China's contact with cradles of zebu**
**domesticates from two directions: overland through Tibet and Yunnan to the southwest**

**and by sea from Southeast Asia to its eastern coast at least 2.9 kya, which overlapped**
**with the appearance of zebu in South China”**

**I’m unsure what results these conclusions are drawn from and how they relate to *Bos***
***javanucis* introgression. Please clarify.**

***Response:*** *We apologize the previously confused description. This part has been revised as*
*“The *Bos javanicus* species historically ranged throughout the southeast mainland and*
*southern China⁴⁸. We observed that Chinese indicine inherited a ~2.92 % genome component*
*from *Bos javanicus* ancestry at least 2.9 kya, which coincides with the earliest evidence for*
*zebu in China 3 kya^{14,15}. The initial period of introgression is most likely due to the eastward*
*migration of zebu to China⁴⁹”*

**[Comment of Reviewer #3:]**

**Supplementary Note 1:**

**Line 26: “sequencing coverage was approximately 11.67X (ranging from 8.87 to 36.85)”**

**Given the variable coverage, it is important to check that missing heterozygosity in**
**lower coverage samples is not affecting downstream analyses such as PCA,**
**ADMIXTURE and NJ trees. Therefore I suggest analysis are repeated that where each**
**sample is made pseudo-haploid to ensure no bias in the results. Alternatively, GL-aware**
**analyses such as Ngsadmix should be applied that take into account genotype**
**uncertainty.**

***Response:*** *Thanks for your suggestions. We have repeated the PCA and ADMIXTURE results*
*by using the genotype likelihood approach, such as PCA-ANGSD and Ngsadmix. The results*
*have been added in the Supplementary Fig. 3 and Supplementary Fig.5. The PCA result was*
*consistent with the previous result. The population structures of K=2 to 5 of two approaches*
*were similar.*

**[Comment of Reviewer #3:]**

**Line 45: “Ascertained in other species within Bovini and phased”**

**Similarly, I potentially worry about the effect of imputing and phasing the lower**
**coverage samples.**

**Response:** Thanks for your comments. This part has been revised “The whole genome data
from 7 extant wild species were mapped in the same way. We used 60.4 million SNPs as
reference list to genotype the combine set of 260 samples and 7 extant wild species. The final
genotype data were imputed and phased using BEAGLE (version 4.1).”

Regarding the imputed data, in our study, we only used imputed data for MSMC
analysis and RFMix analysis. For MSMC analysis, all individuals with deep-coverage (15.72
X to 35.85 X) were selected for analysis.

To detect the effect of imputed data on banteng introgression analysis, we first
extracted the missing genotypes in the introgression region of Chinese indicine group. Then
we calculated the proportion of missing genotypes in specific SNPs in different Bos genus
species(Supplementary Table 29). The result showed that the missing genotypes have lower
proportion in Specific SNPs of different Bos genus. So imputing the lower coverage samples
have little effect on RFMix analysis.

**Supplementary Table 29. Proportion of missing alleles in specific SNPs of different Bos genus**
**species.**

Sample	Species	Missing alleles	Specific SNPs	Missing alleles in Specific SNPs	Proportion of missing alleles in Specific SNPs
Banteng1	Bos javanicus	2,385,779	4,551,280	175,760	0.038618
Banteng2	Bos javanicus	1,629,512	4,551,280	20,771	0.004564
Bision02	Bison bision	415,949	2,460,180	3,490	0.001419
Bision01	Bison bision	2,306,628	2,460,180	152,341	0.061923
Wisent01	Bison bonasus	1,457,280	2,477,513	84,589	0.034143
Wisent02	Bison bonasus	434,223	2,477,513	10,486	0.004232
Wisent03	Bison bonasus	534,416	2,477,513	13,235	0.005342
Yak01	Bos grunniens	623,929	3,092,273	29,831	0.009647
Yak02	Bos grunniens	580,270	3,092,273	26,409	0.00854
Yak03	Bos grunniens	4,135,174	3,092,273	391,329	0.126551
Yak04	Bos grunniens	2,436,909	3,092,273	242,469	0.078411
Yak05	Bos grunniens	1,332,805	3,092,273	112,896	0.036509
Yak06	Bos grunniens	1,018,264	3,092,273	78,608	0.025421
Yak07	Bos grunniens	1,341,774	3,092,273	113,086	0.036571
Yak08	Bos grunniens	1,742,468	3,092,273	159,483	0.051575
Yak09	Bos grunniens	1,243,638	3,092,273	103,064	0.03333
Yak10	Bos grunniens	573,124	3,092,273	26,325	0.008513
Yak11	Bos grunniens	5,717,308	3,092,273	466,308	0.150798
Yak12	Bos grunniens	935,004	3,092,273	40,851	0.013211
Yak13	Bos grunniens	551,396	3,092,273	27,053	0.008749

Gaurus01	Bos gaurus	6,281,380	3,999,129	576,628	0.144188
Gaurus02	Bos gaurus	2,798,378	3,999,129	109,727	0.027438
Buffalo01	Bubalus bubalis	4,543,003	2,241,071	37,147	0.016576
Buffalo02	Bubalus bubalis	4,546,969	2,241,071	38,238	0.017062

**[Comment of Reviewer #3:]**

**Line 124: “We used BEAGLE to infer the haplotype phase and impute missing alleles”**

**Why is phasing being performed for the Y chromosome?**

*Response: Thanks for your comments. We have a wrong description and we did not do the*
 *phasing for Y chromosome and we only use BEAGLE to impute the small number of missing*
 *alleles for Y chromosome. Thus we have revised corresponding sentence. It has been revised*
 *as “We used BEAGLE to impute missing alleles”.*

**[Comment of Reviewer #3:]**

**Line 214: “used software KING to estimate kinship coefficients between all the**

**215 individuals”**

**Why was this done? In addition, KING is not likely to perform well with such low**
 **coverage genomes.**

*Response: Thank you for your comments. We have a wrong description. We only used KING*
 *software to evaluate and check the general relatedness of samples within five “core” group,*
 *which did not lead to any results. Following your suggestion, we have deleted this sentence.*

**[Comment of Reviewer #3:]**

**Line 221: “To further investigate the gene flow between Shimao cattle and different**

**worldwide populations, we removed "all LD" using the --indep-pairwise 50 5 0.2 option**

**in PLINK. The f3 statistics for (Gir; Ancient, population B) were quantified for a set of**

**40 worldwide populations using ~27 M SNPs.”**

**This is a large number of SNPs given SNPs in LD was removed. Please check.**

*Response: We apologize for our mistake. We recalculated the f3 statistic using all autosomal*
 *SNPs. This part has been revised as “To further investigate the gene flow between Shimao*
 *cattle and different worldwide populations, the f3 statistics for (Zebu; Ancient, population B)*

were quantified for a set of 42 worldwide populations using ~50 M SNPs." Result were
consistent with previous result.

**[Comment of Reviewer #3:]**

**Line 248: "The nucleotide diversity (π) of all groups were calculated using a sliding**
**window approach"**

**How were difference in coverages accounted for?**

**Response:** Thank you for the comment. To reduce the effects of different coverage on analysis,
we measured nucleotide diversity at the population level again using ANGSD. For autosomal
chromosomes, we estimated the SFS with ANGSD (-doSaf2). We calculated the π with
-doThetas and the result was consistent with the previous results (Supplementary Fig. 13).
The result also recapitulated that Chinese indicine genomes showed the highest nucleotide
diversity. This part of Supplementary note Note 5 has been revised as "We measured
nucleotide diversity at group level again using genotype likelihood approach. The nucleotide
diversity (π) were measured at the population level. For autosomal chromosomes, we
estimated the SFS with ANGSD (-doSaf2). The nucleotide diversity were calculated with
-doThetas."

**[Comment of Reviewer #3:]**

**Supplementary Figure 7: The 5' 3' damage patterns are very messy. Why are they higher**
**at the 3' position? They are also not very smooth, most plots are smoother than this with**
**a nice exponential decay pattern. Please repeat this analysis using MapDamage so we**
**can see the difference between C>T and G>A changes. I do not know what the colors**
**being show here represent.**

**Response:** Thanks for your suggestion. We have double-checked the raw data and re-plot the
5' and 3' damage pattern. In our previous work, the "--collapse" option was used to merge the
overlapping pair end reads when removing adapter from raw fastq file using AdapterRemoval,
so the mapdamage did not identify the 5' and 3' end correctly in our previous setting of
parameters. Therefore, we recounted the mismatch using ANGSD with "-doMisMatch" option,
and updated the damage pattern.

**Supplementary Figure 10. Nucleotide mis-incorporation patterns at 5'- and 3'- read termini**
 **for eight ancient samples.** Nucleotide mis-incorporation patterns along the first and last 25
 read positions obtained for the eight Shimao cattle before trimming and rescaling. All
 libraries were blunt-ended libraries (New England Biolabs) amplified with AmpliTaq Gold
 DNA polymerase (Life Technologies). Mis-incorporation frequencies are shown for the first
 and last 25 nucleotides of the reads aligned to the bovine reference nuclear genome
 *Btau_5.0.1*. The x-axis provides read positions relative to the read starts (positive numbers)
 and read ends (negative numbers). Red: C to T substitutions; Blue: G to A substitutions; Grey:
 All other substitutions.

**List of the updated tables and figures**

Current version	Previous version
Supplementary Fig. 1	Supplementary Fig. 2
Supplementary Fig. 2	Supplementary Fig. 1
Supplementary Fig. 3	Newly added
Supplementary Fig. 4	Newly added
Supplementary Fig. 5	Supplementary Fig. 3
Supplementary Fig. 6	Newly added
Supplementary Fig. 7	Supplementary Fig. 4
Supplementary Fig. 8	Supplementary Fig. 5
Supplementary Fig. 9	Supplementary Fig. 6
Supplementary Fig. 10	Supplementary Fig.7
Supplementary Fig. 11	Supplementary Fig. 10
Supplementary Fig. 12	Supplementary Fig. 11
Supplementary Fig. 13	Newly added
Supplementary Fig. 14	Supplementary Fig. 12
Supplementary Fig. 15	Supplementary Fig. 13
Supplementary Fig. 16	Supplementary Fig. 14
Supplementary Fig. 17	Supplementary Fig. 15
Supplementary Fig. 18	Supplementary Fig. 16
Supplementary Fig. 19	Newly added
Supplementary Fig. 20	Newly added
Supplementary Fig. 21	Supplementary Fig. 17
Supplementary Fig. 22	Supplementary Fig. 18
Supplementary Table 1	Supplementary Table 1
Supplementary Table 2	Supplementary Table 2
Supplementary Table 3	Supplementary Table 3
Supplementary Table 4	Supplementary Table 4
Supplementary Table 5	Supplementary Table 5
Supplementary Table 6	Supplementary Table 6
Supplementary Table 7	Newly added
Supplementary Table 8	Supplementary Table 7
Supplementary Table 9	Supplementary Table 8
Supplementary Table 10	Supplementary Table 9
Supplementary Table 11	Supplementary Table 10
Supplementary Table 12	Supplementary Table 11
Supplementary Table 13	Supplementary Table 12
Supplementary Table 14	Supplementary Table 13
Supplementary Table 15	Supplementary Table 14
Supplementary Table 16	Supplementary Table 15
Supplementary Table 17	Supplementary Table 16

Supplementary Table 18	Supplementary Table 17	821
Supplementary Table 19	Supplementary Table 18	
Supplementary Table 20	Supplementary Table 19	822
Supplementary Table 21	Supplementary Table 20	823
Supplementary Table 22	Supplementary Table 21	
Supplementary Table 23	Supplementary Table 22	824
Supplementary Table 24	Newly	825
Supplementary Table 25	Newly	
Supplementary Table 26	Supplementary Table 23	826
Supplementary Table 27	Supplementary Table 24	827
Supplementary Table 28	Newly	
Supplementary Table 29	Supplementary Table 25	
Supplementary Table 30	Newly	
Supplementary Table 31	Newly	
Supplementary Table 32	Newly	

Reviewers' comments:

Reviewer #1 (Remarks to the Author):

I'm satisfied with the revised version. The authors have done a good job with several valuable complementary analyses in order to address my concerns. There is still a minor problem with numbering the supplementary material (No supplementary tables 7 and 9 in the supplementary file).

I have assessed the answers to Reviewer #2 's remarks. In my opinion they are satisfactory and the revision addresses the concerns in an appropriate way.

1. "Banteng introgression into Chinese indicine previously reported" correctly addressed

2. "Divergence date between Indian and Chinese indicine cattle"

The new dadi model fits much better with observed data and the estimated N_e are in accordance with the MCMC analysis.

However, in supplementary note 6, I do not understand the sentence beginning line 395 " So we prepared the control data that masked the introgression regions of banteng and yak in Chinese indicine and Tibetan taurine cattle, respectively."

3. "discussion of male-mediated gene flow".

The sentence that has been revised is clear. Do the authors interpret the discordance between paternal and maternal lineages by the moving of females, as suggested by Reviewer 2 ? Is there any other argument to support this hypothesis ?

4. "population structure of Eurasian taurines"

I'm satisfied with the answer.

5. Additional comments were appropriately addressed.

Reviewer #3 (Remarks to the Author):

Most of my technical issues from the previous draft have been corrected. My only concern is the clumsy English, especially for the newly added text (both in the main and supplementary). Overall the new additions seem rushed. The manuscript could do with some proof reading. The new figures are much better however. A few minor issues are identified below

Line 98: "Three zebu breeds (Gir, Brahman, and Nelore) of Indian origin were used in this study, and were imported from India to the Americas approximately 200 years ago"
So are these on the map or not?

Line 100: "A total of 60.04 million autosomal SNPs were identified and used in subsequent analysis"

I still do not know why this is important. I suggest to remove it.

Line 113: "when $K = 5$, the five ancestral components were geographically distributed indicating to European taurine, Eurasian taurine, East Asian taurine, Chinese indicine, and Indian indicine components"

Language a bit clumsy.

Line 125: "Cattle breeds from the Middle East, Africa, Northwest China, North-Central China, and

Southwest China all showed partly hybridization of taurine and indicine (Fig. 2a and 2b)."
Language again a bit clumsy.

Line 172: "The Tibetan cattle were divergent from the other East Asian cattle earlier or undergone a stronger drift after the separation might result the positive D scores in Tibetan cattle."
Clumsy language and what is the context of "positive D scores"? This only has meaning if the reader knows the tree used in D-statistic test.

Line 185: "This increased unique diversity could have been caused by hybridization with different bovine species or introgression events"

And simply large N_e . The frequency distribution of this private variation is important. Are these singletons (so new, suggesting a recent expansion) or common? At a minimum, point to a population expansion being another explanation for this extra diversity.

Figure 5 legend: "zebu- and banteng-specific genotypes"
Specific with regards to each other or to other species?

Line 218: "we next exploited the gene content of 1,852 introgressed intervals that was shared at least two investigated haplotypes"

I don't understand this sentence. Clarify.

Line 281: "Results also confirmed a decline of N_e following the onset of domestication"
Ambiguous. What is this (i.e. "Results") referring to exactly?

Line 674: "To detect introgressions from *Bos javanicus*, we first identified *Bos javanicus*-specific alternate alleles that appeared in *Bos javanicus* and were absent from other domesticated taurine and indicine cattle"

I assume Chinese Indicine were not included in this initial screening for private variants?

From rebuttal:

Line 267: "Our results also revealed that introgression cannot explain the difference in divergence time estimates with MSMC (Supplementary Note 6 and Supplementary Fig. 17), which was consistent with the result for modern humans")

It is completely unclear what this statement is referring to above. Please clarify. It seems a to not follow from any results in the sentence above.

Response: Apologies for the ambiguous sentence. In this part, we wonder that banteng introgression might result in changes in the estimated N_e and divergence time. So we prepared the control data that masked the introgression regions of banteng and yak in Chinese indicine and Tibetan taurine, respectively. The analysis of the control data showed that the patterns of divergence time among different group and population size history did not change. So we concluded that limited introgressions have no impact on demographic history using MSMC. It has been revised as "We first evaluated the impact of introgression on the estimated N_e and divergence time and our results revealed that limited introgression have no impact on demographic history simulation using MSMC."

I think you should specifically mention the masking of introgressed segments in the main text, otherwise what you did to "evaluate introgression" is completely ambiguous to the reader?

Response to Reviewers' comments

Reviewer #1 (Remarks to the Author):

I'm satisfied with the revised version. The authors have done a good job with several valuable complementary analyses in order to address my concerns. There is still a minor problem with numbering the supplementary material (No supplementary tables 7 and 9 in the supplementary file).

Response: We thank the reviewer for the kind words, and taking the time to provide feedback. We have re-uploaded Supplementary tables 7 and 9 on the system as Supplementary Data 1 and Data 2, because they are Excel files and are too large to convert to PDF, so we did not merge them into the Supplementary Information.

I have assessed the answers to Reviewer #2 's remarks. In my opinion they are satisfactory and the revision addresses the concerns in an appropriate way.

Response: Thank you for valuable comments.

1. "Banteng introgression into Chinese indicine previously reported"

correctly addressed

Response: Thanks.

2. "Divergence date between Indian and Chinese indicine cattle"

The new dadi model fits much better with observed data and the estimated N_e are in accordance with the MCMC analysis.

However, in supplementary note 6, I do not understand the sentence beginning line 395 " So we prepared the control data that masked the introgression regions of banteng and yak in Chinese indicine and Tibetan taurine cattle, respectively."

Response: In this part, we want to evaluate the impact of introgression on the estimates of effective population size (N_e) and divergence time in MSMC. Sorry for the unclear sentence. We have added more details in Supplementary note 6 as "We also assessed the impact of introgression on the estimates of effective population size (N_e) and divergence time in MSMC.

We repeated the MSMC analysis using the same data but excluding the genomic regions representing banteng introgression in two Chinese indicine cattle (~3.5 %) and yak to two Tibetan taurine cattle (~1.3 %), respectively (Supplementary Tables 20 and 26). The results showed that the limited introgression did not change the estimates of divergence time and N_e too much (Supplementary Fig. 21)."

3. "discussion of male-mediated gene flow".

The sentence that has been revised is clear. Do the authors interpret the discordance between paternal and maternal lineages by the moving of females, as suggested by Reviewer 2 ? Is there any other argument to support this hypothesis ?

Response: We have looked at this issue again. It is clear that paternal (Y) lineage distribution concords well with autosomal distributions of ancestral components estimated using ADMIXTURE and we make this point. However, our presentation of mtDNA data does not suffice to make a firm declaration of difference – this is not distinct enough and also any difference may not be reliably ascribed to male focused breeding practice rather than drift acting on a single segregating locus. Therefore we now delete the reference to mtDNA difference and simply state, line 316 : "Our results show that paternal lineages have a clear phylogeographical structure which concords with autosomal ancestral components."

4. "population structure of Eurasian taurines"

I'm satisfied with the answer.

Response: Thanks.

5. Additional comments were appropriately addressed.

Response: Thanks.

Reviewer #3 (Remarks to the Author):

Most of my technical issues from the previous draft have been corrected. My only concern is the clumsy English, especially for the newly added text (both in the main and

supplementary). Overall the new additions seem rushed. The manuscript could do with some proof reading. The new figures are much better however. A few minor issues are identified below

Response: Thank you for time and valuable comments.

Line 98: “Three zebu breeds (Gir, Brahman, and Nelore) of Indian origin were used in this study, and were imported from India to the Americas approximately 200 years ago”

So are these on the map or not?

Response: Apologize for the confusing description. In our study, a total of six zebu breeds were used. Yes, all six breeds were on the map. We have revised this sentence as “Gir, Brahman and Nelore, which were imported from India to the Americas approximately 200 years ago¹⁷, were used to represent Indian zebras in this study (Supplementary Note 1).” The Fig. 1 legend has been revised as “Geographic map indicating the origins of the cattle breeds in this study.”

Line 100: “A total of 60.04 million autosomal SNPs were identified and used in subsequent analysis”

I still do not know why this is important. I suggest to remove it.

Response: Thanks for comment. It has been deleted as suggested.

Line 113: “when $K = 5$, the five ancestral components were geographically distributed indicating to European taurine, Eurasian taurine, East Asian taurine, Chinese indicine, and Indian indicine components”

Language a bit clumsy.

Response: This has been revised as “When $K = 5$, we observed five geographically distributed ancestral components labelled: European taurine, Eurasian taurine, East Asian taurine, Chinese indicine, and Indian indicine.”

Line 125: “Cattle breeds from the Middle East, Africa, Northwest China, North-Central China, and Southwest China all showed partly hybridization of taurine and indicine (Fig. 2a and 2b).”

Language again a bit clumsy.

*Response: It has been revised as “Cattle breeds from other regions (Middle East, Africa, Northwest China, North-Central China, and Southwest China) show evidence of hybridization between *Bos taurus* and *Bos indicus*.”*

Line 172: “The Tibetan cattle were divergent from the other East Asian cattle earlier or undergone a stronger drift after the separation might result the positive D scores in Tibetan cattle.”

Clumsy language and what is the context of “positive D scores”? This only has meaning if the reader knows the tree used in D-statistic test.

Response: We have deleted the “positive D scores” and it has been revised as “Tibetan cattle were perhaps subjected to stronger drift after the separation which distorted their outgroup- f_3 value (Fig. 4b).”

Line 185: “This increased unique diversity could have been caused by hybridization with different bovine species or introgression events”

And simply large N_e . The frequency distribution of this private variation is important. Are these singletons (so new, suggesting a recent expansion) or common? At a minimum, point to a population expansion being another explanation for this extra diversity.

Response: We have rephrased the sentence as “This increased unique diversity could be influenced by particular (but unknown) historical demography such as population expansion but the scale of this unique diversity suggests hybridization with or introgression from different bovine species.”

Figure 5 legend: “zebu- and banteng-specific genotypes”

Specific with regards to each other or to other species?

*Response: Thanks for your comments. We have added more information in the Figure 5 legend “The relative frequencies of *Bos indicus*-specific genotypes (red dots) and *Bos javanicus*-specific genotypes (blue dots). *Bos javanicus*-specific alternate alleles were*

identified as those that appeared in Bos javanicus genomes and were absent from taurine and Indian indicine cattle genomes. The yak-specific genotypes were identified using the same method.”

Line 218: “we next exploited the gene content of 1,852 introgressed intervals that was shared at least two investigated haplotypes ”

I don’t understand this sentence. Clarify.

Response: Apologize for the ambiguous sentence. It has been revised as “we next exploited the gene content of 1,852 introgressed intervals that were shared by at least two haplotypes in the Chinese indicine group”

Line 281: “Results also confirmed a decline of N_e following the onset of domestication”

Ambiguous. What is this (i.e. “Results”) referring to exactly?

Response: Thanks for comment. The results refer to the decline of N_e after domestication summarized in Figure 6a. It has been revised as “We also observed a decline of N_e during 7 to 9 kya consistent with the onset of domestication (Fig. 6a).”

Line 674: “To detect introgressions from Bos javanicus, we first identified Bos javanicus-specific alternate alleles that appeared in Bos javanicus and were absent from other domesticated taurine and indicine cattle”

I assume Chinese Indicine were not included in this initial screening for private variants?

Response: We apologize for the confusing description. Yes, Chinese indicine were not included. It has been revised as “To detect introgressions from Bos javanicus, we first identified Bos javanicus-specific alternate alleles that appeared in Bos javanicus and were absent from other domesticated taurine and Indian indicine cattle.”

From rebuttal:

Line 267: “Our results also revealed that introgression cannot explain the difference in divergence time estimates with MSMC (Supplementary Note 6 and Supplementary Fig. 17), which was consistent with the result for modern humans”)

It is completely unclear what this statement is referring to above. Please clarify. It seems a to not follow from any results in the sentence above.

I think you should specifically mention the masking of introgressed segments in the main text, otherwise what you did to “evaluate introgression” is completely ambiguous to the reader?

Response: Apologies for the ambiguous sentence. In this part, we wonder that banteng introgression might result in changes in the estimated N_e and divergence time. So we prepared data that masked the introgression regions of banteng and yak in Chinese indicine and Tibetan taurine, respectively.

We now state in the main text line 275: “We applied this method to all groups with two deep-coverage (>15 X) individuals per group. To evaluate the impact of introgression on the estimates of effective population size (N_e) and divergence time, we repeated the MSMC analysis using the same data but excluding the introgressed regions from banteng to two Chinese indicine cattle (~3.5 %) and yak to two Tibetan taurine cattle (~1.3 %) (Supplementary Tables 20 and 26), respectively. The results showed that the limited introgression did not notably change the estimates of divergence time and N_e (Supplementary Fig.21)